# Gut barrier defects, intestinal immune hyperactivation and enhanced lipid catabolism drive lethality in NGLY1-deficient *Drosophila*

Ashutosh Pandey [1] ✉, Antonio Galeone[2,7], Seung Yeop Han[1], Benjamin A. Story[3], Gaia Consonni[2], William F. Mueller[3], Lars M. Steinmetz[3,4], Thomas Vaccari [2] & Hamed Jafar-Nejad [1,5,6] ✉

Intestinal barrier dysfunction leads to inflammation and associated metabolic changes. However, the relative impact of gut bacteria versus non-bacterial insults on animal health in the context of barrier dysfunction is not well understood. Here, we establish that loss of *Drosophila N*-glycanase 1 (Pngl) in a specific intestinal cell type leads to gut barrier defects, causing starvation and JNK overactivation. These abnormalities, along with loss of *Pngl* in enterocytes and fat body, result in Foxo overactivation, leading to hyperactive innate immune response and lipid catabolism and thereby contributing to lethality. Germ-free rearing of *Pngl* mutants rescued their developmental delay but not lethality. However, raising *Pngl* mutants on isocaloric, fat-rich diets partially rescued lethality. Our data indicate that Pngl functions in *Drosophila* larvae to establish the gut barrier, and that the lethality caused by loss of *Pngl* is primarily mediated through non-bacterial induction of immune and metabolic abnormalities.

Intestinal barrier dysfunction allows various pathogens and non-living stimuli to induce the innate immune response[1–5]. Although the goal of innate immune response induction is to restore intestinal homeostasis, its hyperactivation can have local and systemic adverse effects and is implicated in the pathogenesis of human diseases including autoimmune and neurodegenerative disorders[6–8]. In addition, hyperactive immune response can be accompanied by profound changes in metabolism including high energy demand and subsequent depletion of the nutrient depot[9,10]. For example, induction of chronic inflammation results in the aggravation of metabolic disorders in mouse models of obesity[11]. However, the relative contribution of infectious versus non-infectious mechanisms to detrimental consequences caused by gut barrier dysfunction is not well understood.

The gut mucus layer is one of the key components regulating intestinal barrier functions[12]. Equivalent to the mammalian gut mucus layer, a peritrophic matrix (PM) is present in the *Drosophila* intestine[13]. The PM is composed of highly glycosylated proteins and chitin, and is continuously secreted from a group of specialized cells called peritrophic matrix-forming ring (PR) cells in the proventriculus region at the junction of foregut and midgut[13,14]. There is strong evidence that in addition to chitins and mucin-type *O*-glycans, PM also contains *N*-glycoproteins[15,16]. Moreover, lectin-based studies in another insect suggest that *N*-glycoproteins might control the functional properties

[1]Department of Molecular & Human Genetics, Baylor College of Medicine, Houston, USA. [2]Department of Biosciences, University of Milan, Milan, Italy. [3]Genome Biology Unit, European Molecular Biology Laboratory (EMBL), Heidelberg, Germany. [4]Department of Genetics, School of Medicine, Stanford University, Stanford, USA. [5]Genetics & Genomic Graduate Program, Baylor College of Medicine, Houston, USA. [6]Development, Disease Models & Therapeutics Graduate Program, Baylor College of Medicine, Houston, USA. [7]Present address: Institute of Nanotechnology, National Research Council (CNR-NANOTEC), Lecce, Italy. ✉e-mail: Ashutosh.Pandey@bcm.edu; hamedj@bcm.edu

of PM like its permeability[17]. However, genetic evidence for the contribution of *N*-glycoproteins to gut barrier function in *Drosophila* is lacking.

Terminally misfolded proteins are detected by the endoplasmic reticulum-associated degradation (ERAD) pathway and retro-translocated from ER to the cytosol for proteasomal degradation[18]. One of the major branches of ERAD uses *N*-glycans as a signal to recognize misfolded *N*-glycoproteins and shuttles them to the retro-translocation machinery[18]. The cytosolic deglycosylating enzyme *N*-glycanase 1 (NGLY1) removes *N*-glycans from misfolded proteins and is thought to function in the ERAD pathway[19–21]. Recessive mutations in human *NGLY1* cause a congenital disorder of deglycosylation named NGLY1 deficiency[22–24]. It is an ultra-rare disorder that leads to global developmental delay and affects multiple organ systems including the nervous system and the gastrointestinal system. Loss of the *Drosophila* homolog of human NGLY1 (PNGase-like or Pngl) results in semi-lethality, as less than 1% of homozygous mutant animals finish the larval and pupal development and eclose as an adult organism[25,26]. We have previously reported that loss of *Pngl* in the visceral mesoderm impairs signaling pathways mediated by decapentaplegic (Dpp; homolog of human bone morphogenetic protein 4, BMP4) and adenosine monophosphate-activated protein kinase (AMPK) in the larval intestine, which leads to structural and functional intestinal phenotypes and contributes to the lethality of *Pngl*[−/−] animals[26,27]. However, impaired BMP and AMPK signaling due to mesodermal loss of *Pngl* did not fully explain the lethality of *Pngl* mutants, suggesting critical roles for Pngl in other biological processes and potentially in other cell types.

Here, we report that loss of *Pngl* leads to increased expression of the innate immune genes in the *Drosophila* larval intestine and a systemic increase in lipid catabolism, compromising developmental progression and leading to lethality. We find that loss of *Pngl* in the PR cells leads to abnormalities in peritrophic matrix and impairment in the gut barrier function. Our data suggest that the gut barrier defects result in increased activation of Foxo in the gut epithelial cells, both via enhanced stress-induced JNK signaling and through a systemic starvation response. In addition, we observe that Pngl is required cell-autonomously in enterocytes and in the fat body to prevent aberrant Foxo activation and to repress lipid catabolism. Importantly, while germ-free rearing does not rescue the lethality of *Pngl* mutants, increasing the lipid content in isocaloric diets improves their survival to adulthood, suggesting that the mutant animals lack sufficient energy stores to reach the adult stage. Altogether, our data suggest that Pngl is required to establish the gut barrier in *Drosophila* larvae and that the lethality associated with gut barrier defects in *Pngl* mutants is primarily caused by non-bacterial insults.

## Results

### *Pngl*-mutant midguts show upregulation of immune genes

Animals homozygous for a *Pngl* null allele (*Pngl*[ex14/ex14], *Pngl*[−/−] hereafter) display severe developmental delay, with the majority of mutant animals not reaching the pupal stage[26]. Therefore, in this study, we analyzed age-matched third instar larvae. To determine the biological processes that contribute to lethality in *Pngl* mutants, we performed transcriptomic analysis using RNA sequencing (RNA-seq) on a mixed pool of male and female third instar larval midguts of *Pngl*[−/−] animals and three control strains: *y w* (*Pngl*[+/+]), *Pngl*[+/−], and *Pngl*[−/−]; *Pngl Dp*/+, which lacks endogenous *Pngl* function but harbors one copy of a *Pngl* genomic duplication shown to fully rescue the lethality of *Pngl*[−/−] animals[27]. Principal component analysis (PCA) showed that the first two principal components drive the majority of the variance among the samples, and that all samples show strong distinctive clustering by genotype (Fig. 1a). We then identified the genes differentially expressed between *Pngl*[−/−] and each control, as those showing an absolute fold-change of at least 1.5 and a False Discovery Rate (FDR) equal to or

less than 0.05 (Supplementary Data 1). We generated a Shiny app that can be used to perform pairwise and three-way comparisons of these datasets and to download the corresponding differentially expressed gene lists (Supplementary Data 2; https://shiny-portal.embl.de/shinyapps/app/14_flyvenn). To increase the stringency of our analysis, we focused on those differentially expressed genes that were overlapping among these three pairwise comparisons: (1) *Pngl*[−/−] vs *y w*, (2) *Pngl*[−/−] vs *Pngl*[+/−], and (3) *Pngl*[−/−] vs *Pngl*[−/−]; *Pngl Dp*/+. We found 459 upregulated and 469 downregulated genes in *Pngl*[−/−] larval midguts, when compared to all controls (Fig. 1b and Supplementary Data 3). Using the Database for Annotation, Visualization, and Integrated Discovery (DAVID)[28,29], we performed functional gene ontology (GO) analysis on the differentially expressed genes and identified various biological processes significantly altered in *Pngl*[−/−] midguts (Fig. 1c and Supplementary Data 4). We found proteasome-mediated processes as the topmost significantly downregulated gene category (Fig. 1c; top panel), in agreement with previous reports on the regulation of proteasomal gene expression by *Pngl* and its homologs[30–34].

Interestingly, a number of significantly upregulated gene categories were related to immune response (Fig. 1c; bottom panel). Many of the upregulated genes in the immune response gene categories encode for antimicrobial peptides (AMPs) and pattern recognition receptor proteins such as peptidoglycan recognition proteins (PGRPs). qRT-PCR analysis on 13 innate immune response genes from the list confirmed the RNA-seq results (Fig. 1d). Addition of a genomic copy of *Pngl* (*Pngl Dp*) significantly reduced the immune gene expression (Fig. 1d). We conclude that loss of *Pngl* leads to a significant increase in the expression of multiple immune response-related genes in the larval midgut. Importantly, impaired proteasomal gene expression or AMPKα signaling, both of which have been shown to be affected in *Pngl*-deficient animals[27,33], cannot explain the severe increase in the expression of innate immunity genes in *Pngl*[−/−] larval midguts (Supplementary Figs. 1a, 1b and 2a).

### Enhanced midgut immune response contributes to *Pngl*[−/−] lethality

Expression of AMPs and other innate immune genes in *Drosophila* is regulated by two signaling pathways, the Toll pathway and the immune deficiency (IMD) pathway[35]. Moreover, in the intestinal epithelium, the forkhead transcription factor Foxo can induce the expression of AMPs in response to starvation, energy deprivation, and infection[36,37]. To determine if the increased immune gene expression contributes to the lethality of *Pngl* mutants, we decreased the gene dosage of *foxo*, *Rel* (encodes Relish, which is the NF-κB transcription factor in the IMD pathway) and *Tl* (encodes the Toll receptor). Loss of one copy of each gene in *Pngl*[−/−] larvae resulted in a significant reduction in the midgut expression of the immune response genes, although in most cases the expression levels did not fully return to control levels (Fig. 2a). Notably, we observed a more robust rescue upon reducing *foxo* compared to *Rel* and *Tl* (Fig. 2a). Reducing the gene dosage of *foxo* rescued the lethality of *Pngl* mutants by 40%, while decreasing the gene dosage of *Rel* and *Tl* rescued the lethality of *Pngl*[−/−] animals by 19% and 21%, respectively (Fig. 2b). Moreover, combined heterozygosity for *foxo* and *Rel* or *Tl* in *Pngl*[−/−] animals did not further increase the degree of lethality rescue achieved by reducing *foxo* gene dosage alone (Fig. 2b). These observations suggest that these genes contribute to *Pngl*[−/−] lethality through a common mechanism, likely the induction of innate immune genes. Together, these observations suggest that *foxo*-mediated hyperactivation of innate immune genes is a major contributor to the lethality in *Pngl* mutants.

The *Drosophila* larval midgut epithelium is composed of enterocytes, enteroendocrine cells and adult midgut precursor cells[38]. *Pngl* knockdown in enterocytes, but not enteroendocrine or adult midgut precursor cells, led to partial lethality and immune gene induction (Fig. 2c, d). Enterocyte-specific knockdown of *foxo*,

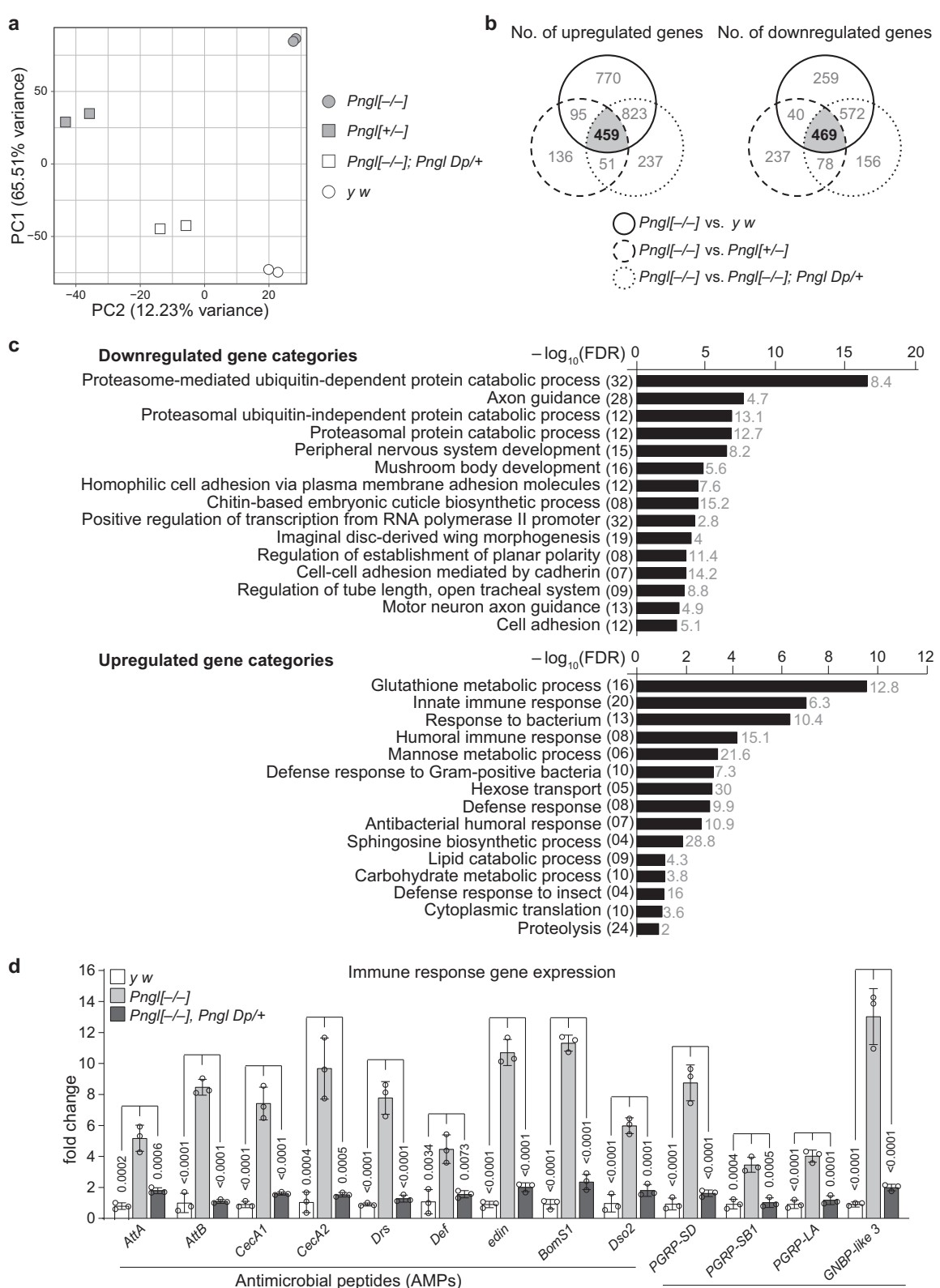

**Fig. 1 | Loss of Pngl is associated with the upregulation of immune response-related genes. a** Principal component analysis (PCA) plot showing the variance among the genotypes used in RNA-seq. **b** Venn diagrams showing the overlap of differentially regulated genes in *Pngl*[−/−] with comparison to control (*y w*), *Pngl*[+/−] and *Pngl*[−/−], *Pngl Dp*/+ in RNA-seq analysis. Number of both upregulated and down-regulated genes shown in the Venn diagram were based on >1.5 fold-change. **c** Graph presents DAVID functional GO analysis of biological processes (BP) of downregulated (top), and upregulated (bottom) genes based on their −log₁₀ of

FDR. Numbers in parenthesis show the number of genes differentially expressed in each category and the numbers next to bars show the fold enrichment. **d** Graph showing expression of immune response-related genes in the indicated genotypes. Values are expressed as fold changes relative to control (*y w*). Mean ± standard deviation of three independent replicates is shown. Numbers on the bars indicate the *P* values. Significance is ascribed as *P* < 0.05 using one-way ANOVA with multiple comparisons followed by Šidák correction. Source data are provided as Source Data file.

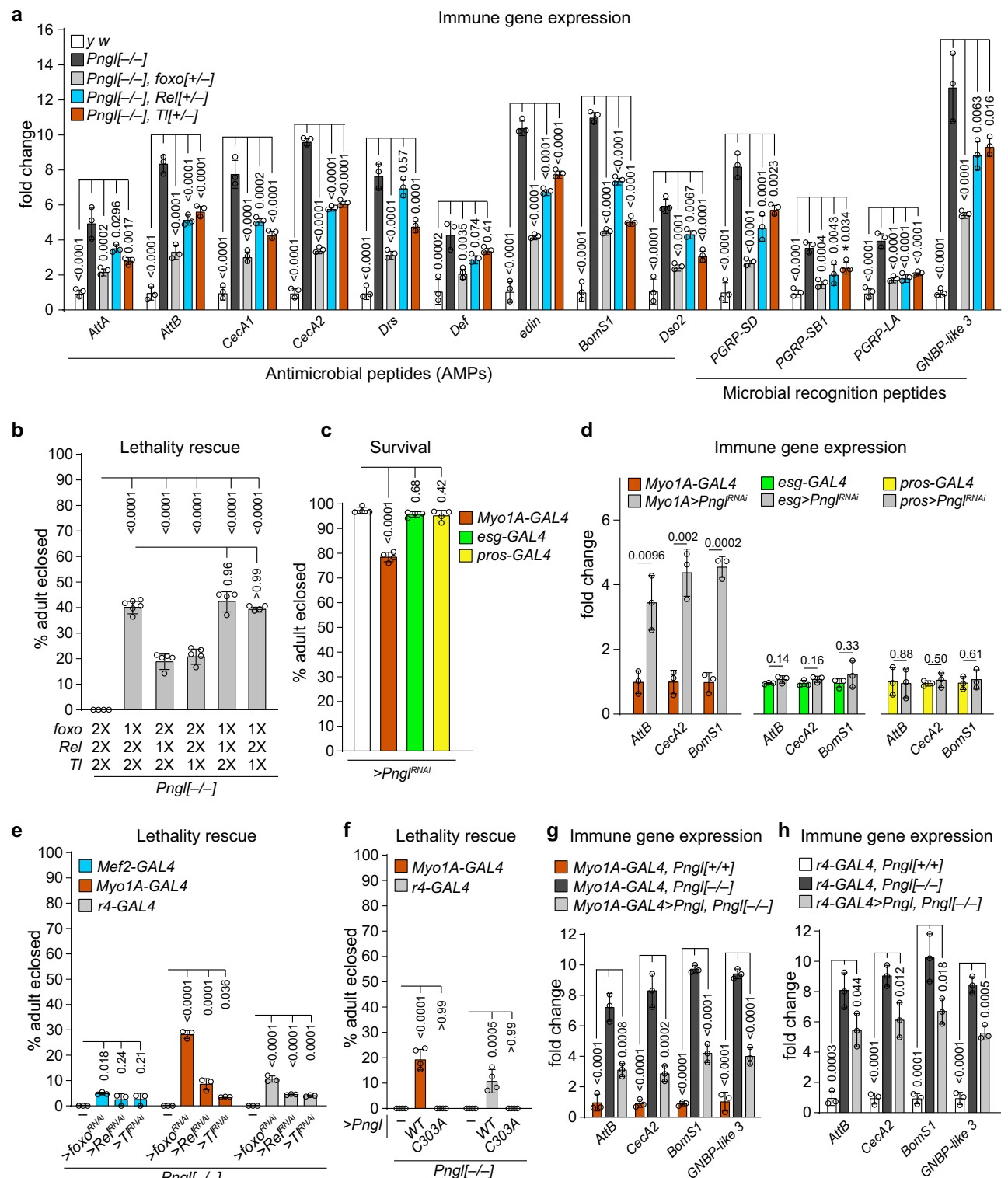

**Fig. 2 | Enhanced innate immune response in midgut contributes to the lethality of *Pngl* mutants. a** Graph showing immune response gene expression (relative fold change to control, *y w*) in the midgut of age-matched larvae of the indicated genotypes (*n* = 3 independent replicate). **b** Graph showing % lethality rescue in *Pngl* mutants upon removing one copy (1X) of each immune gene activator (*n* = 4–6 independent replicates per genotype). **c** Graph showing survival upon *Pngl* knockdown driven by the indicated GAL4 drivers (*n* = 4 independent replicates). **d** Graph showing immune gene expression (relative fold change to genetic control) in the indicated genotypes (*n* = 3 independent replicates). **e** Graph showing % lethality rescue in *Pngl* mutants upon tissue-specific knockdown of

immune activators (*foxo*, *Rel* and *Tl*) using the indicated GAL4 drivers (*n* = 3 independent replicates). **f** Graph showing % lethality rescue in *Pngl* mutants upon overexpression of wild-type (*Pngl^WT*) and catalytically inactivate (*Pngl^C303A*) *Pngl* using indicated GAL4 drivers (*n* = 4 independent replicates). **g**, **h** Graphs showing immune gene expression in the midgut of age-matched larvae of indicated genotypes (*n* = 3 independent replicates). In all panels, each circle represents an independent replicate, and mean ± standard deviation is shown. Numbers on bars indicate the *P* values. Significance is ascribed as *P* < 0.05 using one-way ANOVA with multiple comparisons followed by Šidák correction in (**a**–**c**, **e**–**h**) or two-tailed unpaired t-test (**d**). Source data are provided as Source Data file.

*Rel,* and *Tl* resulted in a lethality rescue of ~28%, 8%, and 3% in *Pngl* mutants, respectively (Fig. 2e). We also included the fat body in our analysis, given its prominent role in the systemic release of AMPs[39]. Fat body-specific knockdown of *foxo, Rel*, and *Tl* also led to modest but statistically significant rescue of lethality (Fig. 2e). Taken together, these data suggest that the detrimental effects of immune hyperactivation in *Pngl* mutants primarily results from Foxo-mediated induction of immune response genes in enterocytes and to some extent in the fat body.

Overexpression of wild-type *Pngl* (*Pngl^WT*) but not a catalytically-inactive form of *Pngl* (*Pngl^C303A*) using *Myo1A-GAL4* and *r4-GAL4* drivers led to ~19% and ~11% rescue of *Pngl^-/-* lethality, respectively (Fig. 2f). These observations indicate that in addition to its role in mesoderm[26], the enzymatic activity of Pngl is also required in enterocytes and fat body during *Drosophila* development. Notably, enterocyte- and fat body-specific overexpression of *Pngl* in *Pngl* mutants led to a significant reduction of innate immune gene expression in the midgut but did not fully rescue this phenotype (Fig. 2g, h). These data suggest that the hyperactive immune response in *Pngl* mutant midguts is due to the loss of *Pngl* in several cell types.

### *Pngl^-/-* larvae exhibit Foxo overactivation in intestine and fat body

Foxo is negatively regulated by insulin receptor (InR)/Akt signaling[40]. Under conditions of starvation or energy deprivation, decrease in insulin signaling allows the nuclear localization of Foxo and subsequent induction of its target genes[41]. We found increased nuclear localization of Foxo and a significant reduction in phospho-Foxo (pFoxo) levels in *Pngl^-/-* midguts, which were rescued by *Pngl Dp* (Fig 3a, b), suggesting increased Foxo activation in midgut upon loss of *Pngl*. Western blot analysis showed a significant decrease in pAkt levels in *Pngl^-/-* midguts (Fig. 3c). Moreover, we observed a lethality rescue of ~11% and ~17% by enterocyte-specific overexpression of *Akt^WT* and constitutively active Akt (*Akt^CA*), respectively, in *Pngl* mutants (Fig. 3d). Furthermore, overexpression of *Akt^CA* in enterocytes led to a significant reduction in innate immune gene expression in *Pngl* mutants (Fig. 3e). Western blot analysis showed that enterocyte-specific overexpression of *Pngl* rescues pAkt levels in *Pngl^-/-* midguts (Fig. 3f). Together, these data suggest that reduced Akt activation in *Pngl^-/-* enterocytes contributes to Foxo overactivation and increased immune gene expression in the midgut, and lethality.

Small body size is a signature of decreased insulin signaling in flies[42]. *Pngl^-/-* larvae showed growth retardation compared to controls (Fig. 3g). In agreement with a previous report[43], we observed a small body size phenotype in *Pngl^-/-* third instar larvae, which was rescued by decreasing *foxo* (Fig. 3h, i). Accordingly, our data suggest that reduced insulin signaling in the midgut might contribute to *Pngl^-/-* lethality. The *Drosophila* insulin-like peptide 6 (Dilp6; official name Ilp6) is primarily expressed in third instar larval fat body and is essential for animal growth[44]. However, *dilp6* overexpression in the fat body did not rescue the *Pngl^-/-* lethality (Fig. 3j). We also overexpressed another insulin-like peptide (Dilp2) in the fat bodies of *Pngl* mutants but did not see any lethality rescue (Fig. 3j). Overexpression of *dipl2* and *dilp6* did not improve the small body size phenotype either (Fig. 3k). We next examined InR expression in midguts and found that it is significantly decreased in *Pngl^-/-* midguts (Fig. 3l). Overexpression of *InR^WT* and constitutively activate InR (*InR^CA*) in enterocytes led to a modest but statistically significant rescue of the *Pngl^-/-* lethality (Fig. 3m). A lower degree of rescue was also observed upon *InR^WT* and *InR^CA* overexpression in the fat body (Fig. 3m). Overexpression of *InR^WT* and *InR^CA* in enterocytes and fat body also exhibited a partial rescue of body size in *Pngl* mutants (Fig. 3n). These observations suggest that reduced InR signaling in enterocytes and potentially fat body contributes to the lethality of *Pngl* mutants.

Reduced InR signaling can result from starvation[45]. One of the hallmarks of starvation in *Drosophila* larvae is an increase in lipid catabolism. Importantly, one of the upregulated gene categories in the RNA-seq analysis was related to lipid catabolic process (Fig. 1c; bottom panel and Fig. 3o), which was confirmed by qRT-PCR experiments (Fig. 3p). These observations suggest that starvation contributes to reduced InR signaling and Foxo activation in *Pngl* mutant larvae.

### *Pngl^-/-* larvae exhibit gut barrier defects

Gut barrier defects can lead to activation of the intestinal innate immune response[2,3]. To examine whether *Pngl^-/-* larvae exhibited any gut barrier dysfunction, we fed the control and *Pngl*-mutant larvae on food containing FITC-labeled high molecular weight (500-kDa) dextran. In control midguts, the FITC signal was restricted to the central parts of the lumen area, but the *Pngl^-/-* midguts failed to retain the FITC signal in the lumen area (Fig. 4a, b). Around 55% of the examined *Pngl*-mutant third instar larvae (but none of the WT) showed gut barrier defect, and providing one genomic copy of *Pngl* fully rescued the phenotype (Fig. 4a, c). *Pngl^-/-* larvae show a gut clearance defect due to reduced *AMPKα* expression in visceral mesoderm[27]. However, adding an *AMPKα Dp* did not rescue the gut barrier defect in *Pngl^-/-* larvae (Supplementary Fig. 2b). These observations indicate that Pngl is required for normal gut barrier formation in *Drosophila* larvae independently of the AMPKα defects previously reported in these animals.

Given the impaired gut barrier function of *Pngl^-/-* larvae, we examined the integrity of peritrophic matrix (PM) by marking the PM in *Pngl^-/-* and control larvae with Helix pomatia agglutinin (HPA) lectin, which selectively binds to α-*N*-acetylgalactosamine residues and is specific for *O*-glycans[46,47]. In control larvae, PM separated a central luminal area from a peripheral "peritrophic" space adjacent to the apical surface of the midgut epithelium (Fig. 4d), an arrangement which is thought to ensure that abrasive food particles and microorganisms pass through the gut without contacting the epithelial cells[13]. However, although PM can clearly be observed in the midgut of *Pngl^-/-* larvae, it is highly disorganized and appears to be collapsed on itself (Fig. 4d). This is accompanied by epithelial irregularities, as evidenced by defects in apical phalloidin staining and detachment of some epithelial cells (Fig. 4d). In addition, analysis of midgut sections by light microscopy showed the lumen area lined with intact PM and a peritrophic space between enterocytes and PM in control animals. In contrast, *Pngl^-/-* midguts displayed a dense lumen content (potentially due to the gut clearance defect previously reported in these animals[27]), disorganized PM, and loss of peritrophic space (Fig. 4e). These observations indicate that impaired gut barrier function in *Pngl^-/-* larvae is associated with PM abnormalities.

To determine if loss of *Pngl* in PR cells (present in proventriculus) contributes to the gut barrier defects observed in *Pngl^-/-* larvae, we overexpressed *Pngl* in PR cells of *Pngl* mutants using *path-GAL4* and fed the larvae with 500-kDa FITC-dextran. We found that the penetrance of gut barrier phenotype in *Pngl^-/-* larvae was reduced from 55% to ~20% upon PR-specific overexpression of *Pngl^WT*, but not *Pngl^C303A* (Fig. 4f). This was accompanied by a lethality rescue of ~11% (Fig. 4g). In addition to proventriculus, *path-GAL4* also drives expression in larval fat body, brain and salivary glands (Supplementary Fig 3a). However, *Pngl* overexpression in these tissues with specific drivers did not rescue the gut barrier defect of *Pngl^-/-* larvae (Supplementary Fig. 3b). Additionally, *Pngl* knockdown driven by *path-GAL4*, but not by other drivers, recapitulates the gut barrier defect (Supplementary Fig 3c). Together, these data indicate that the deglycosylation activity of Pngl is required in PR cells to ensure the integrity of the PM and gut barrier and to promote the survival of *Drosophila* larvae.

To assess whether there is an accumulation of *N*-glycoproteins in the PR cells of *Pngl* mutant larvae, we stained the proventricular

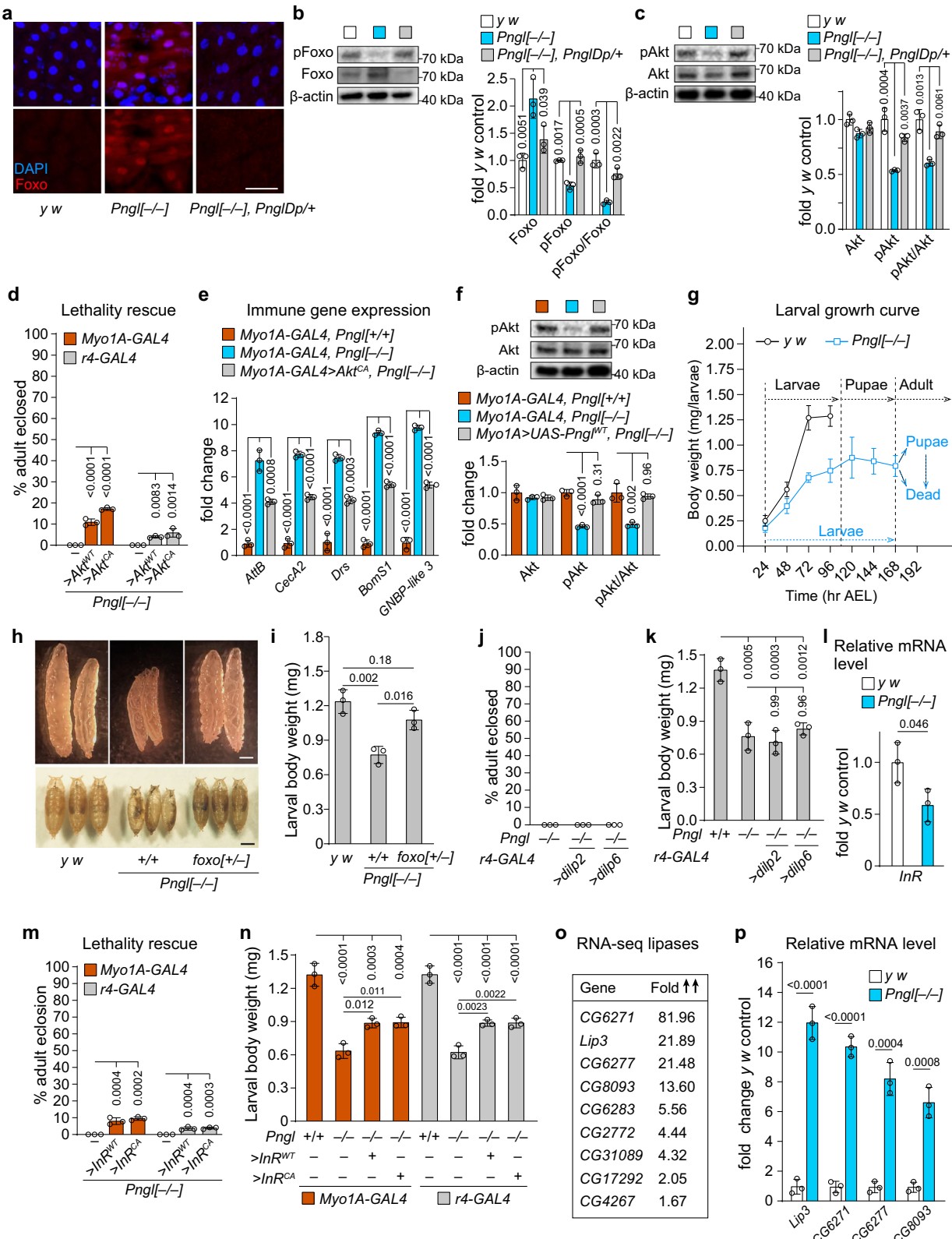

regions of these animals and control third instar larvae with two lectins: wheat germ agglutinin (WGA), which recognizes *N*-acetylglucosamine in *O*-glycans, chitins, and *N*-glycans, and concanavalin A (Con A), which primarily binds high-mannose *N*-glycans[15,48]. Notably, *Pngl* mutants showed a significant increase in the number and size of ConA⁺ intracellular puncta in the PR cells, which was rescued by *Pngl Dp*, suggesting that loss of *Pngl* might affect the trafficking and/or

secretion of some *N*-glycoproteins (Fig. 4h–j). A similar increase was observed in the number of WGA⁺ puncta in *Pngl⁻/⁻* PR cells (Supplementary Fig. 3d, e). Importantly, overexpression of *Pngl^WT*, but not *Pngl^C303A*, in PR cells rescued the number and size of the ConA⁺ intracellular puncta in *Pngl* mutants (Fig. 4k–m), suggesting a key role for the enzymatic activity of Pngl in PR cells. These observations provide a potential mechanism for the PM defects observed in *Pngl⁻/⁻* larvae.

**Fig. 3 | Reduced Akt phosphorylation and enhanced Foxo activation in *Pngl* mutant midguts. a** Confocal images showing DAPI (blue) and Foxo (red) staining in the larval midguts (96 hr after egg laying, AEL) of indicated genotypes. Scale bar is 50 μm. **b** Western blot images and quantification showing the Foxo and pFoxo levels in the larval midguts (96 hr AEL) of indicated genotypes. **c** Western blot images and quantification showing Akt and pAkt level in the larval midgut (96 hr AEL) of indicated genotypes. **d** Graph showing % lethality rescue of *Pngl* mutants in indicated genotypes. *Akt^{WT}*, wild-type *Akt*; *Akt^{CA}*, constitutively active *Akt*. **e** Graph showing immune gene expression in the larval midgut (96 hr AEL) of indicated genotypes. **f** Western blot images and quantification showing Akt and pAkt levels in the larval midgut (96 hr AEL) of indicated genotypes. **g** Graph showing the growth rate curve of control and *Pngl^{-/-}* animals. **h** Images of larvae (96 hr AEL) and pupae of the indicated genotypes. Scale bars are 500 μm. **i** Graph showing quantification of larval body weight (96 hr AEL) in indicated genotypes. **j** Graph showing % lethality rescue of *Pngl* mutants in indicated genotypes. **k** Graph showing quantification of larval body weight (96 hr AEL) in indicated genotypes. **l** Graph showing the expression of *InR* in the midguts of age-matched control and *Pngl^{-/-}* larvae. **m** Graph showing % rescue of *Pngl^{-/-}* lethality in indicated genotypes. *InR^{WT}*, wild-type *InR*; *InR^{CA}*, constitutively active *InR*. **n** Graph showing larval body weight (96 hr AEL) in indicated genotypes. **o** Genes in the "Lipid catabolic process" category from the larval midgut RNA-seq, along with the fold increase in their expression level in *Pngl^{-/-}* compared to *y w* control. **p** Graph showing the expression of lipases in *Pngl^{-/-}* midgut. In all panels, the mean ± standard deviation of three independent replicates is shown. *P* values are indicated on bars. Significance is ascribed as $P < 0.05$ using one-way ANOVA followed by Šidák correction (**b**–**f**, **i**, **k**, **m**, **n**) or two-tailed unpaired student's t-test (**l**, **p**). Source data are provided as Source Data file.

## Germ-free rearing does not rescue the *Pngl^{-/-}* lethality

The gut barrier defects and upregulation of multiple microbial recognition peptides in *Pngl^{-/-}* larvae (Figs. 4 and 1d) suggest that the gut microbiota might contribute to some of the *Pngl^{-/-}* phenotypes. Moreover, altered gut microbiota can precede intestinal barrier defects in aging adult *Drosophila*[49]. To directly examine the role of gut microbiota in *Pngl*-mutant phenotypes, we generated germ-free (GF) *Pngl^{-/-}* and control larvae. GF *Pngl* mutants did not show any improvement in their gut barrier phenotype (Fig. 5a), suggesting that the PM defects caused by loss of *Pngl* are not due to altered gut microbiota.

The GF *Pngl^{-/-}* larvae showed a significant increase in the percentage of larvae that reach the pupal stage compared to conventionally-reared *Pngl^{-/-}* larvae (Fig. 5b), suggesting an important role for gut microbiome in the developmental delay of *Pngl* mutants. However, GF rearing did not lead to any rescue in the *Pngl^{-/-}* lethality (Fig. 5c). Of note, GF rearing of *Pngl^{-/-}; foxo^{+/-}* larvae resulted in further improvement in their survival compared to conventionally-reared *Pngl^{-/-}; foxo^{+/-}* animals (Fig. 5c, also compare to Fig. 2b) suggesting that the contribution of gut microbiome to *Pngl^{-/-}* lethality is redundant to that of Foxo hyperactivation. Expression of the genes involved in the recognition of microorganisms is not statistically different between GF *Pngl^{-/-}* and GF *y w* larval midguts (Fig. 5d). However, the GF *Pngl^{-/-}* larval midguts still exhibit a significant increase in the expression of most AMPs (Fig. 5d). Together, these data suggest that although gut microbiota plays an important role in the developmental delay of *Pngl^{-/-}* larvae, it is not the major inducer of immune gene expression and lethality in these animals.

## Gut barrier defects lead to starvation and *Pngl^{-/-}* lethality

To gain mechanistic insight into the role of gut barrier defects in the lethality of *Pngl^{-/-}* larvae and to separate the effects of gut barrier defect from the cell-autonomous effects of loss of *Pngl* in the enterocytes and fat body, we induced gut barrier defects in control (*Pngl^{+/+}*) animals by feeding them with polyoxin D (Poly D), a chitin synthase inhibitor that disrupts PM formation in insects[50,51]. Poly D feeding for 48 hours resulted in a 43%-penetrant gut barrier phenotype in *y w* animals (Fig. 6a), confirming that this strategy can impair PM formation in *Drosophila* larvae. Poly D feeding also led to ~19% lethality, and increased expression of immune genes and lipases (Fig. 6b–d), indicating gut barrier defect can cause immune gene induction, starvation and lethality in *Drosophila* larvae.

Poly D-fed *foxo^{+/-}* larvae showed a 38%-penetrant gut barrier phenotype, indicating that as expected, loss of one copy of *foxo* does not affect the ability of Poly D to impair gut barrier integrity (Fig. 6a). However, removing one copy of *foxo* rescued the lethality caused Poly D feeding and significantly reduced the expression of most immune genes induced by this chemical (Fig. 6b, c). Further, western blot analysis revealed a significant reduction in the relative levels of pFoxo in midguts of Poly D-fed larvae, suggesting Foxo activation (Fig. 6e). Importantly, quantification of the number of mandibular movements

per second suggests a comparable feeding behavior in control and *Pngl*-mutant larvae (Supplementary Fig. 4). Therefore, the starvation signature observed in these animals is not likely to be due to reduced feeding. Together, these observations indicate that gut barrier defects can lead to a starvation-like condition in *Drosophila* larvae and suggest that Foxo likely operates downstream of this starvation-like phenotype to mediate the lethality associated with gut barrier defects.

Overexpression of InR or Akt in the midgut was less effective in rescuing the *Pngl^{-/-}* lethality compared to *foxo* knockdown by using the same driver (compare Fig. 3d, m to Fig. 2f). Therefore, in addition to reduced InR signaling, other pathways are likely to contribute to Foxo activation and lethality in *Pngl^{-/-}* midguts. The Jun-N-terminal Kinase (JNK) pathway is a potential candidate for these effects, as it is activated in epithelial tissues by a variety of extrinsic and intrinsic stressors and can induce the nuclear localization and activation of Foxo[52,53]. Indeed, feeding Poly D to control larvae led to the activation of JNK signaling in larval midgut, as evidenced by the accumulation of phospho-JNK (pJNK) in their midgut epithelial cells (Fig. 6f). Western blot analysis showed that enterocyte-specific knockdown of the *Drosophila* JNK homolog *basket* (*bsk*) suppressed the Foxo activation in the midgut of Poly D-fed larvae (Fig. 6e). *Pngl^{-/-}* larvae also accumulated pJNK in their midgut epithelial cells (Fig. 6g). Moreover, *bsk* knockdown in enterocytes led to a lethality rescue of 22% in *Pngl^{-/-}* larvae (Fig. 6h), accompanied by a significant decrease in immune gene expression and a significant decrease in Foxo nuclear localization in midguts (Fig. 6i, j). Altogether, these observations suggest that gut barrier defects lead to increased JNK signaling in *Pngl^{-/-}* midgut epithelial cells, which itself contributes to Foxo activation and lethality in these animals.

## *Pngl* mutants exhibit an increase in lipid catabolism

Starvation can lead to lipid mobilization[45,54]. In line with the significant increase in the expression of multiple lipase genes in *Pngl^{-/-}* midguts (Fig. 3o, p), we observed a significant reduction in the lipid storage in fat body and midgut of *Pngl* mutants (Fig. 7a). We also found a significant decrease in triacylglycerol (TAG) level in hemolymph and midgut of *Pngl* mutants throughout the third instar larval period (Fig. 7b), accompanied by a significant increase in free fatty acid levels in hemolymph of early third instar *Pngl* mutants (Fig. 7c, 72 hr). By mid-third instar larval stage, free fatty acid levels return to normal (Fig. 7c, 108 hr), potentially suggesting the depletion of energy reserve as the development of *Pngl^{-/-}* larvae proceeds. Loss of one copy of *foxo* improved all of these phenotypes (Fig. 7a–c). Together, these data indicate that *Pngl^{-/-}* larvae experience a significant degree of starvation associated with a Foxo-dependent increase in lipid catabolism.

## Dietary lipid supplementation partially rescues the *Pngl^{-/-}* lethality

In the standard fly food used in our experiments, only 2% of the total energy contents is provided by lipids. The depletion of lipid storage in *Pngl^{-/-}* midgut and fat body prompted us to examine whether

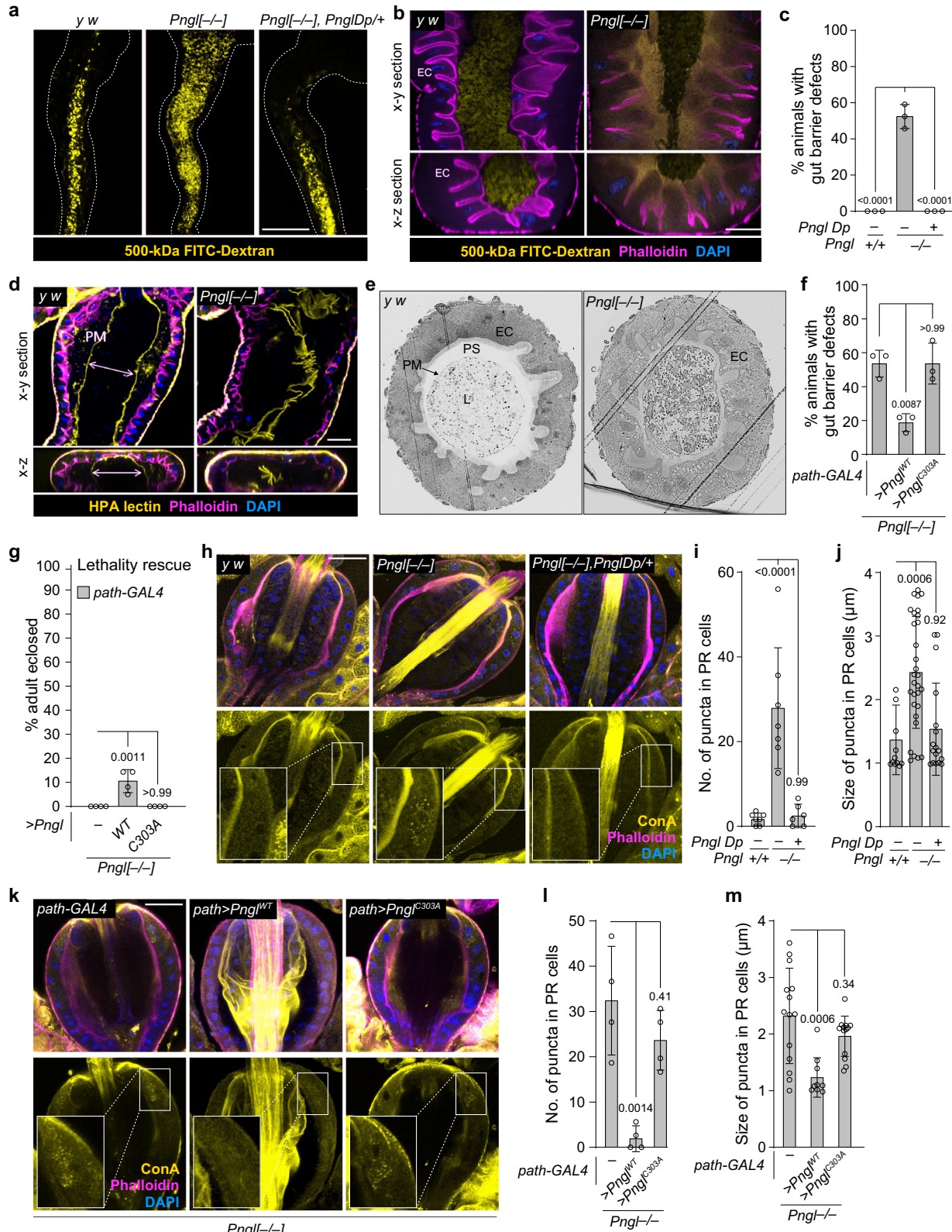

increasing the fat content of the food can promote the survival of *Pngl*[−/−] larvae. To this end, we used two additional isocaloric food compositions with different fat contents in our experiments: a "high-protein intermediate-fat diet" (HPIFD) and a "high-fat diet" HFD (Fig. 7d and Supplementary Table 1). On our standard diet (SD), on average 19.6% of *Pngl*[−/−] larvae reached the pupal stage (Fig. 7e). However, when grown on isocaloric HPIFD and HFD, 37.3% and 77.2% of *Pngl*[−/−] larvae

reached the pupal stage, respectively (Fig. 7e). Further analysis indicated that the developmental delay of *Pngl*[−/−] larvae was partially rescued by HPIFD and fully rescued by HFD (Supplementary Fig. 5a). Remarkably, HPIFD and HFD also led to *Pngl*[−/−] lethality rescues of 19.2% and 41.1%, respectively (Fig. 7f). Immunostaining indicated that HFD significantly reduced the Foxo nuclear localization in midgut and fat body of *Pngl* mutants (Supplementary Fig. 5b), and partially

**Fig. 4 | Loss of Pngl is associated with gut barrier defect which contributes to the lethality. a** Low magnification fluorescent images of the midguts of indicated genotypes upon FITC-labeled dextran (yellow) feeding. **b** Confocal images showing phalloidin (magenta) and DAPI (blue) staining in the indicated genotypes upon FITC-labeled dextran (yellow) feeding. **c** Graph showing the quantification of gut barrier defect phenotype in the indicated genotypes (*n* = 60 for each genotype from three independent experiments). **d** Confocal images showing phalloidin (magenta), DAPI (blue), and HPA-lectin (yellow) staining for peritrophic matrix in control and *Pngl* mutant midguts. **e** Light microscopic images of cross-section of control and *Pngl* mutant midguts. L, lumen; PM, peritrophic matrix; PS, peritrophic space; EC, enterocytes. **f** Graph showing quantification of gut barrier defect phenotype in the indicated genotypes (*n* = 60 for each genotype from three independent experiments). **g** Graph showing % lethality rescue in the indicated genotypes. **h** Confocal images showing Phalloidin (magenta), DAPI (blue), and ConA (yellow) lectin staining in the proventriculus region of third instar larvae of indicated

genotypes (*n* = 6–7 for each genotype). Insets show close-up of PM-secreting cells. **i** Graph showing the quantification of the number of intracellular puncta in the indicated genotypes. **j** Graph showing the size of intracellular puncta in the indicated genotypes (*n* = 12–30 puncta from three animals per genotype). **k** Confocal images showing Phalloidin (magenta), DAPI (blue), and ConA (yellow) lectin staining in the proventriculus region of third instar larvae of indicated genotypes (*n* = 4 for each genotype). Insets show a close-up of PM-secreting cells. **l** Graph showing the quantification of the number of intracellular puncta in the indicated genotypes. **m** Graph showing the size of intracellular puncta in the indicated genotypes (*n* = 9–14 puncta from four animals per genotype). The scale bar is 250 μm in (**a**) and 50 μm in (**b**, **d**, **h**, **k**). In all panels, each circle represents an independent replicate in (**c**, **f**, **g**, **i**) or a punctum in (**j**, **m**) and mean ± standard deviation is shown. *P* values are indicated on bars. Significance is ascribed as *P* < 0.05 using one-way ANOVA followed by Šidák correction. Source data are provided as Source Data file.

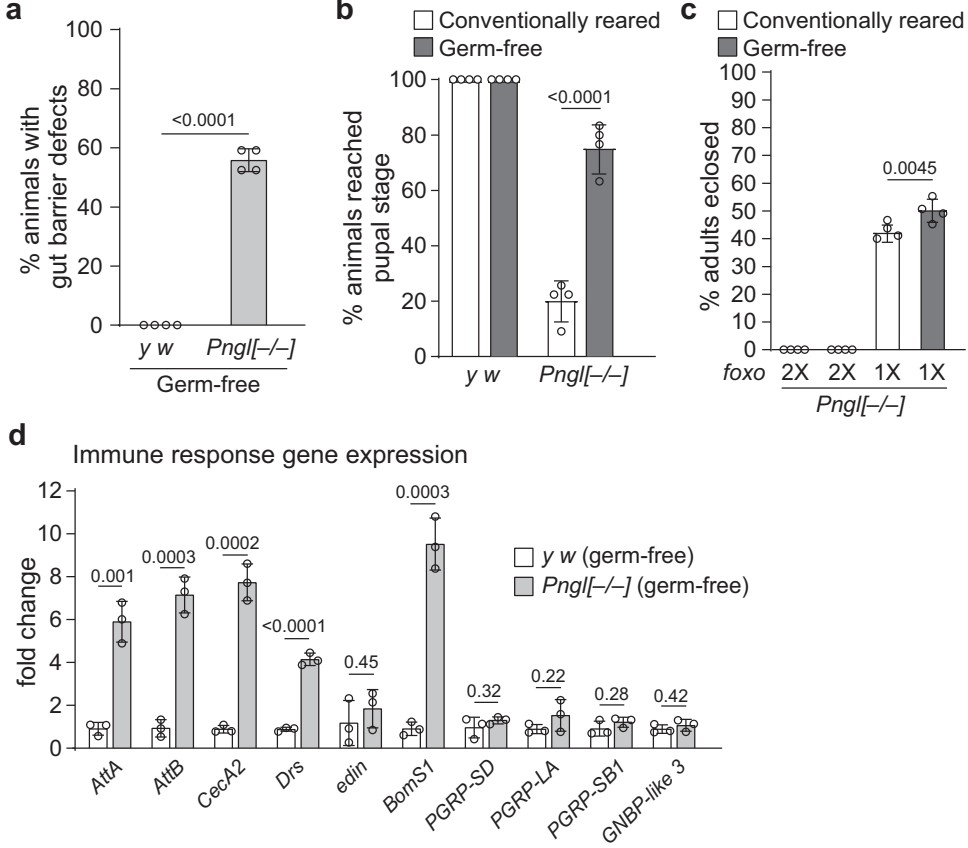

**Fig. 5 | Gut microbiota does not explain gut barrier defects and lethality in *Pngl* mutants. a** Graph showing quantification of gut barrier defect phenotype in germ-free control and *Pngl*[−/−] larvae (60 animals for each group from four independent experiments). **b** Graph showing the percentage of germ-free and conventionally reared control and *Pngl*[−/−] animals that reached the pupal stage (120 animals for each group from four independent experiments). **c** Graph showing % lethality rescue in germ-free and conventionally reared *Pngl*[−/−] and *Pngl*[−/−]; *foxo*[+/−] animals

(*n* = 4 independent replicates). **d** Graph showing immune gene expression in germ-free control and *Pngl*[−/−] larvae (*n* = 3 independent replicates). In all panels, each circle represents an independent replicate, and mean ± standard deviation is shown. *P* values are indicated on bars. Significance is ascribed as *P* < 0.05 using two-way ANOVA followed by Šidák correction (**b**, **c**) or two-tailed unpaired student's t-test (**a**, **d**). Source data are provided as Source Data file.

rescued innate immune gene overexpression in *Pngl* mutants (Supplementary Fig. 5c). These data underscore the contribution of metabolic abnormalities and energy depletion to the developmental delay and lethality in *Pngl* mutants.

## Discussion

We previously reported that Pngl is required in the midgut visceral mesoderm for the regulation of Dpp and AMP kinase signaling in *Drosophila*[26,27,55]. However, impairment of these two pathways only partially explained the lethality of *Pngl*[−/−] animals. Here, we report that

*Pngl* plays critical roles in several other cell types (PR cells, enterocytes, and fat body cells). Loss of *Pngl* leads to gut barrier defects, as well as suppression of InR signaling and activation of JNK signaling in enterocytes. These alterations result in overactivation of Foxo in enterocytes and fat body, which in turn leads to hyperactivation of intestinal innate immune genes and increased lipid catabolism, ultimately causing lethality (Fig. 7g). *Pngl* mutants show significant developmental delay and less than 1% of the animals reach adulthood[25,26]. Intriguingly, growing *Pngl* mutants on an isocaloric high-fat diet fully rescued their developmental delay and allowed ~41% of the animals to

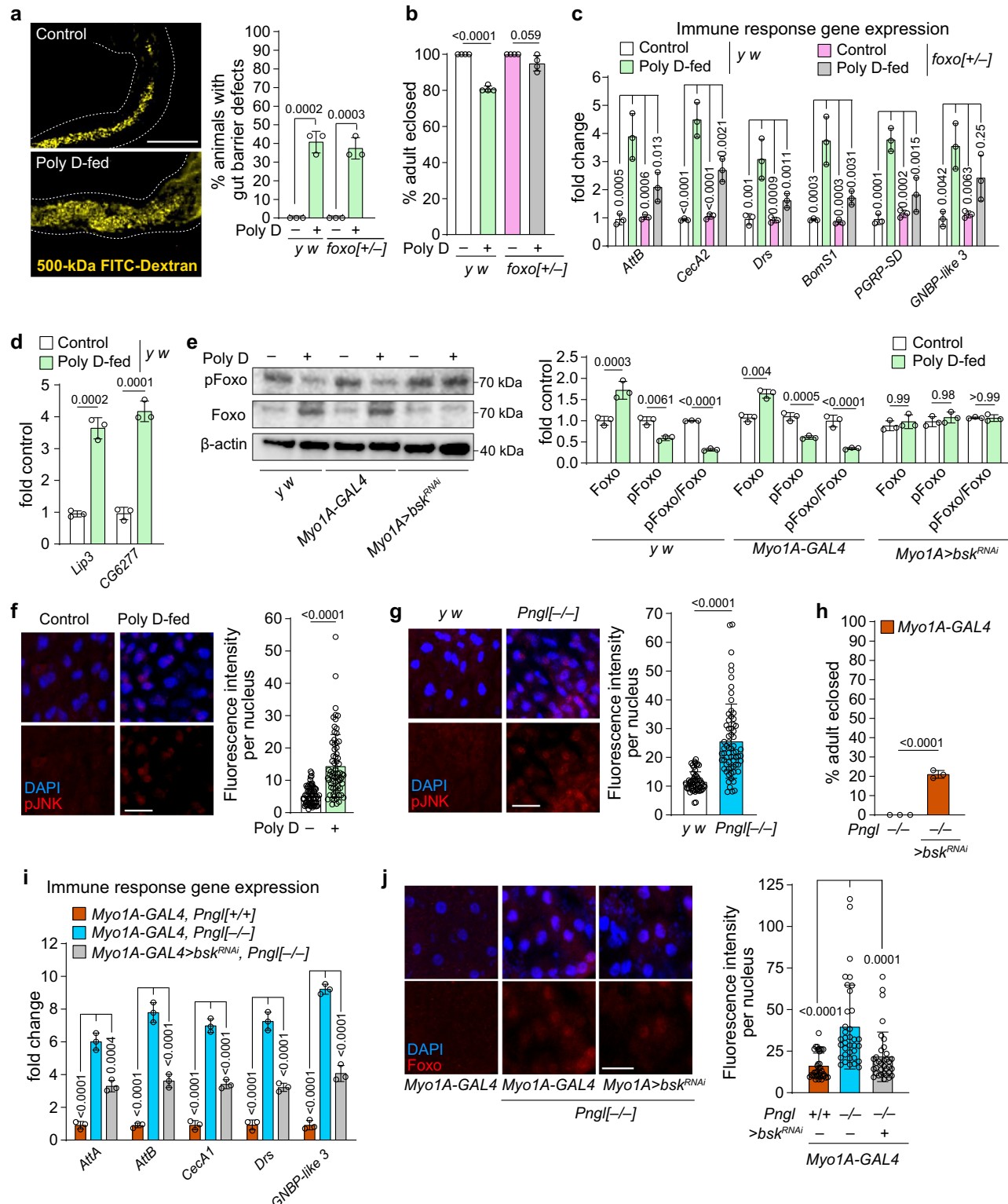

reach adulthood, strongly suggesting that depletion of energy depot during the larval and pupal development plays a critical role in the lethality of *Pngl* mutants. We note that a high-fat diet also suppressed Foxo overactivation in the fat body and midgut of *Pngl*[-/-] larvae. This observation suggests that by worsening the starvation status, enhanced lipid catabolism engages in a vicious cycle with Foxo over-activation and negatively impacts the survival of *Pngl*[-/-] animals. We note that nutrient deprivation can also suppress lipogenesis in enterocytes and the fat body through enhanced secretion of Tachykinins by

enteroendocrine cells[56]. It remains to be examined whether reduced lipogenesis also contributes to low lipid content in *Pngl*[-/-] larvae.

Our data provide strong evidence that hyperactivation of innate immune genes contributes to lethality in *Pngl* mutants. First, decreasing the gene dosage of *foxo*, *Rel* and *Tl*, which are the known regulators of innate immune gene expression, partially rescued the lethality in *Pngl* mutants. In addition, in all of our other genetic and diet-induced rescue experiments, rescue of lethality was accompanied by reduced immune gene expression. It has recently been reported

**Fig. 6 | Gut barrier defects induce starvation and Foxo-dependent induction of innate immune genes and lethality. a** Confocal images showing FITC-labeled dextan (yellow) in midguts from control and Poly D-fed larvae of indicated genotypes and quantification of gut barrier defect in these animals (*n* = 3 independent replicates). Scale bar is 250 µm. **b** Graph showing % adult eclosion in Poly D-fed control and *foxo*[+/−] animals (*n* = 4 independent replicates). **c** Graph showing immune response gene expression in control and Poly D-fed *y w* and *foxo*[+/−] midguts (*n* = 3 independent replicates). **d** Graph showing gene expression of lipases in control and Poly D-fed midguts of *y w* larvae (*n* = 3 independent replicates). **e** Western blot images and quantification graph showing Foxo and pFoxo levels in the midguts of Poly D-fed larvae of indicated genotypes (*n* = 3 independent replicates). **f** Confocal images showing DAPI (blue) and pJNK (red) staining and quantification in control and Poly D-fed *y w* midguts. The scale bar is 50 µm. *n* = 69 nuclei from four animals for each group. **g** Confocal images showing DAPI (blue) and pJNK (red) staining and quantification in control and *Pngl*[−/−] midguts. The scale bar is 50 µm. n = 56 (*y w*) and 66 (*Pngl*[−/−]) nuclei from four animals for each group. **h** Graph showing % lethality rescue upon enterocyte-specific knockdown of *Drosophila* JNK (*bsk*) in *Pngl* mutants (*n* = 3 independent replicates). **i** Graph showing innate immune gene expression in the indicated genotypes (*n* = 3 independent replicates). **j** Confocal images showing DAPI (blue) and Foxo (red) staining and quantification of Foxo nuclear localization in the indicated genotypes. The scale bar is 50 µm. *n* = 35–36 nuclei from three animals for each group. Each circle represents an independent replicate in (**a–e, h, i**) or a nucleus in (**f, g, j**). Mean ± standard deviation is shown in all graphs. *P* values are indicated on bars. Significance is ascribed as *P* < 0.05 using one-way ANOVA followed by Šidák correction (**e, i, j**), two-way ANOVA followed by Šidák correction (**c**) or two-tailed unpaired student's t-test (**a, b, d, f–h**). Source data are provided as Source Data file.

that overexpression of AMPs results in cytotoxicity and cell death in aging *Drosophila*[7]. Our data suggest immune hyperactivation can also induce lethality during development and that Pngl prevents immune hyperactivation in part by helping establish a normal gut barrier.

Although *foxo* heterozygosity had the biggest rescue effect in *Pngl* mutants, loss of one copy of *Rel* and *Tl* also showed some degree of lethality rescue. Since pathogens are thought to be the main inducers of the Imd and Toll pathways in *Drosophila*[57], we anticipated that gut microbiota contributes to the lethality of *Pngl*[−/−] animals. Rather surprisingly, GF rearing of *Pngl* mutants did not rescue lethality at all, even though it dramatically rescued the developmental delay in these animals. How can one reconcile these observations? Since GF rearing also removes commensal bacteria, one possibility is that commensal bacteria are important for gut homeostasis in *Pngl* mutants. Regardless of the reason for this discrepancy, our data indicate that upon gut barrier defect in *Pngl* mutants, gut microbiota are not the primary mediators of hyperactive immune response.

Accumulation of ConA[+] and WGA[+] puncta in PM-secreting cells in *Pngl*[−/−] larvae suggests that loss of Pngl might affect the secretion of some *N*-glycoproteins. Components of the *Drosophila* larval PM have not been systematically identified. However, studies in other insects indicate that some PM components are *N*-glycoproteins[16,17]. It remains to be studied whether the gut barrier defects observed in *Pngl*[−/−] larvae result from a failure in the deglycosylation and secretion of key *N*-glycoprotein components of the PM. Notably, a number of putative PM components (Muc68Ca, Muc26B, Muc96D, Muc11A, and Muc68E) were downregulated in *Pngl*[−/−] midguts based on our RNA-seq analysis (Supplementary Data 3), suggesting an alternative mechanism for PM abnormalities in *Pngl* mutants.

Human insulin receptors harbors 19 predicted *N*-glycosylation sites, 14 of which have been experimentally verified[58]. Importantly, *Drosophila* InR is predicted to have 12 *N*-glycosylation sites (NetNGlyc 1.0 server prediction[59]). Therefore, while the reduction in InR signaling in *Pngl*[−/−] larvae can be explained by starvation and reduced expression of *InR*, it is also plausible that the InR protein itself is a direct target of Pngl and shows abnormal trafficking and/or function upon loss of Pngl-mediated deglycosylation. Further studies are required to identify and characterize the biologically relevant targets of NGLY1/Pngl in the *Drosophila* larval intestine and fat body.

Human patients with NGLY1 deficiency display an array of symptoms including global developmental delay, lack of tears, and chronic constipation[60]. Some of the NGLY1 deficiency patients were reported to have recurrent, severe respiratory infections, while others were reported to have higher than expected antibody titers against rubella and rubeola after Measles, Mumps, and Rubella (MMR) vaccination[23]. Transcriptome profiling indicated significant upregulation of genes involved in immunity in NGLY1 deficiency patient fibroblasts compared to control fibroblasts[34]. Moreover, global gene profiling in *Ngly1*-deficient melanoma cells showed upregulation of cytokines such as interferon β1 and interleukin 29 (ref. [61]), and *Ngly1*[−/−] mouse embryonic

fibroblasts exhibit increased expression of the interferon genes[31]. In addition, RNA-seq in adult flies with RNAi-medicated *Pngl* knockdown also showed an increase in the expression of the innate immune genes[33]. Although these reports linked NGLY1 deficiency and altered immune response in various contexts, the current study is the first report on the role of NGLY1/Pngl in gut immunity and systemic metabolism in an in vivo model. Of note, altered glycosylation in intestinal epithelial cells is implicated in chronic inflammatory diseases including inflammatory bowel disease[62]. Considering these reports, the critical roles uncovered here for *Drosophila* Pngl in regulating the gut mucus barrier, innate immune response, and metabolic homeostasis warrant further studies to explore whether loss of NGLY1 in other systems or alterations in *N*-glycosylation machinery exert similar immune and metabolic effects.

## Methods

### *Drosophila* strains and culture

Animals were reared at room temperature on standard food containing cornmeal, molasses, and yeast in all experiments except for those involving high-protein, intermediate-fat diet (HPIFD) and high-fat diet (HFD). The detailed composition of these diets is listed in Supplementary Table 1. The following *Drosophila* strains were used in the study: (1) *y w*, (2) *foxo*[A94]/TM6B, Tb[1], (3) *Tl*/TM3, Sb[1], (4) *Rel*[E38], (5) *UAS-foxo*[RNAi], (6) *UAS-Tl*[RNAi], (7) *y*[1] *sc*[*] *v*[1] *sev*[21];*UAS-Rel*[RNAi], (8) *Myo1A-GAL4* (enterocytes), (9) *r4-GAL4* (fat body), (10) *c135-GAL4* (*path-GAL4*), (11) *UAS-dipl2*, (12) *UAS-InR*[WT], (13) *UAS-InR*[AI325D] (constitutively activated *UAS-InR*[CA]), (14) *UAS-Akt*[WT], (15) *UAS-Akt*[ΔPH] (constitutively activated *UAS-Akt*[CA]), (16) *UAS-mCherry*[nls], (17) *pros-GAL4* (enteroendocrine cells), (18) *esg-GAL4* (adult midgut precursors), (19) *elav-GAL4* (neurons), (20) *C147-GAL4* (salivary glands) and (21) *UAS-bsk*[RNAi] (Bloomington Drosophila Stock Center); (22) *Mef2-GAL4* (mesoderm; ref. [63]); (23) *Pngl*[ex14], (24) *UAS-Pngl*[WT] and (25) *UAS-Pngl*[C303A] (ref. [25]); (26) *PBac{Pngl*[wt]*}VK31* (*Pngl* duplication; ref. [27]); (27) *Dp(1;3)DC102*, *PBac{DC102}VK33* (*AMPKα* duplication; ref. [64]); and (28) *UAS-dipl6* (ref. [65]).

### Transcriptomic analysis/RNA sequencing

Midgut tissues from a mixed pool of male and female third instar larvae at 96 hours after egg laying were dissected and homogenized in groups of 25 in a cold solution of Tri-reagent (Sigma-Aldrich, T9424); the amount of RNA in each sample was determined by Nanodrop, and RNA quality was analyzed using agarose gel electrophoresis (1.2%). Two biological replicates were used for each genotype. The samples were prepared on a Beckman FXP using the Illumina TruSeq stranded mRNA chemistry and sequenced on a NextSeq 500 (Mid Output flowcell) in paired-end mode. Raw paired-end sequencing reads were trimmed using cutadapt v1.12 to remove Illumina adapters and low-quality bases. The processed reads were then aligned to the *Drosophila* dm6 genome using STAR v2.5.3a. Gene-level counts, based on GTF annotations from FlyBase Release 6.20, were tabulated by totaling all reads overlapping the collapsed set of exons for each gene following

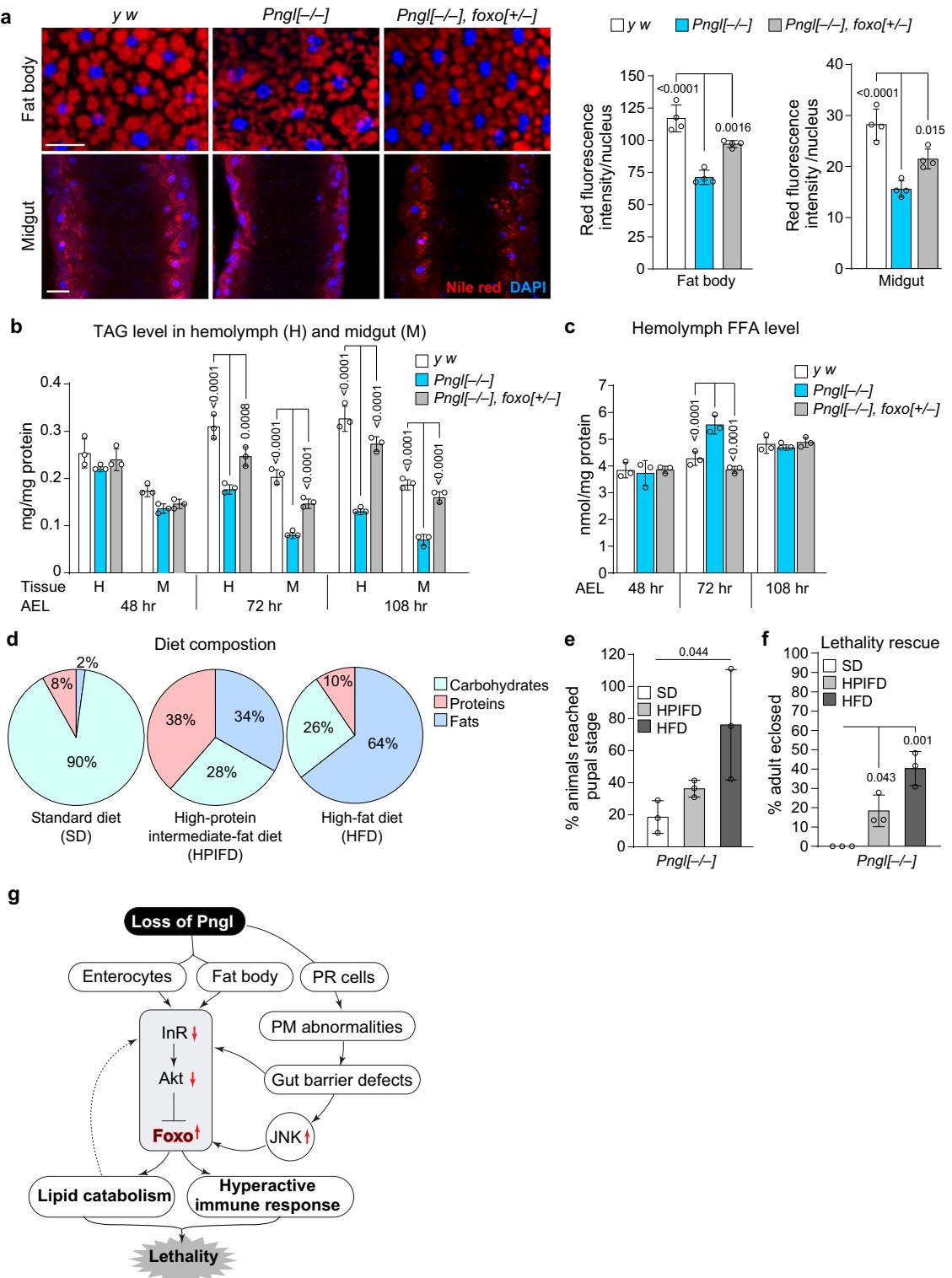

**Fig. 7 | *Pngl* mutants exhibit increased lipid catabolism and supplementation on dietary lipid partially rescues their lethality. a** Images showing Nile red (red) and DAPI (blue) staining in fat body and midgut of the indicated genotypes and their quantification ($n = 4$ independent replicates). Scale bars are 50 μm. **b** Triacylglycerol (TAG) level in hemolymph and midgut of indicated genotypes ($n = 3$ independent replicates). **c** Free fatty acid (FFA) level in the hemolymph of indicated genotypes ($n = 3$ independent replicates). **d** Pie charts showing % carbohydrates, proteins and fats in different diet compositions. **e** Graph showing % animals that reach the pupal stage upon feeding on the indicated diets ($n = 3$

independent replicates). **f** Graph showing lethality rescue in *Pngl* mutants upon feeding on the indicated diets ($n = 3$ independent replicates). **g** Schematic model showing that loss of *Pngl* in several cell types of the *Drosophila* larvae results in Foxo overactivation and subsequent innate immune gene expression and lipid catabolism, leading to lethality. In all panels, each circle represents an independent replicate, and mean ± standard deviation is shown. *P* values are indicated on bars. Significance is ascribed as $P < 0.05$ using one-way ANOVA followed by Šidák correction (**a**, **e**, **f**) or two-way ANOVA followed by Šidák correction (**b**, **c**). Source data are provided as Source Data file.

previously published methods[66]. Genes with a counts per million reads mapped (CPM) of less than 5 in less than two samples were excluded. The principal component analysis (PCA) plot was generated using the log-transformed CPM. Differentially expressed genes were identified by using the edgeR glmFit model. The false discovery rate (FDR) was controlled by applying the Benjamini–Hochberg procedure to the $P$ values. Significantly expressed genes were defined as those exhibiting an absolute fold-change of at least 1.5 and an FDR of less than or equal to 0.05. The raw data have been deposited in the Gene Expression Omnibus (GEO) under accession number GSE206229.

## Developmental assay
For developmental assay, the expected ratio was calculated based on Mendelian inheritance for genotypic classes and the observed/expected ratio is reported as a percentage. Crosses made up of 5 virgin females and 5 males were set in each tube after a period for sexual maturation and housing to obtain the maximum fitness level during the three days of egg deposition in a non-overcrowded environment. Crosses were set in triplicate and flies were transferred to fresh vials every 3 days for three times. The total number of pupae for genotypic classes produced over 20 days in each vial was scored.

## Quantification of lethality rescue
In lethality rescue experiments, the number of genetic siblings and their Mendelian ratio were used to calculate the expected number of the target progeny. Lethality rescue in *Pngl* mutants was examined as described previously[26]. We scored the number of adult progeny including *Pngl* homozygous and heterozygous animals. The number of homozygous animals is expected to be 50% of the number of heterozygous animals. Accordingly, we calculated the expected number of homozygous animals based on the observed number of heterozygous animals. For example, if in a genetic cross, we observe 50 heterozygotes, we would expect 25 homozygotes. Considering this expected homozygous number as 100%, we calculated the % homozygotes eclosed. We can formulate the calculation as % lethality rescue = Observed number of homozygotes x 100 / Observed number of heterozygotes x 0.5. For all genetic crosses, 3-6 independent replicates were used.

## Real time quantitative RT-PCR analysis
Total RNA was extracted from 5 larval midguts with Trizol (Invitrogen) and dissolved in 25 µL of RNase-free water. cDNA was then synthesized from 1 µg total RNA using amfiRivert II cDNA Synthesis Master Mix (R5500, GenDEPOT), and qPCR was carried out using amfiSure qGreen Q-PCR Master Mix, Low ROX (Q5601, GenDEPOT). Expression levels were normalized to *Actin* (endogenous control). Relative gene expression was calculated as fold change using the $2^{-\Delta\Delta Ct}$ method. The oligonucleotides used to assess target genes expression are listed in Supplementary Table 2.

## Dissections, staining, image acquisition, and processing
Larval midgut and fat body tissues were dissected and fixed in 4% paraformaldehyde. Antibodies were rabbit anti-dFoxo 1:250 (ab195977, abcam) and rabbit anti-SAPK/JNK 1:500 (Cat No. 559309, Sigma-Aldrich). Lectin staining in the proventriculus region was performed using helix pomatia agglutinin (HPA), Alexa Fluor™ 488 conjugate 1:1000 (cat No. L11271, Invitrogen), wheat germ agglutinin (WGA) CF®488 A 1:1000 (Cat No. 29022, Biotium), and concanavalin A (ConA) CF®488 A 1:1000 (Cat No. 29016, Biotium). Confocal images were acquired using a Leica TCS-SP8 microscope using LAS X 3.1.5 software and processed with Amira5.2.2. Images were generated as a limited projection view of 5-10 optical sections. Quantifications were performed using ImageJ. All images were processed in Adobe Photoshop CC. Figures were assembled in Adobe Illustrator CC. For comparison between different groups, color intensity per nucleus was calculated in 3-4 independent replicates.

## Western blotting
Protein samples were prepared from larval midguts in lysis buffer containing Halt™ Phosphatase Inhibitor Single-Use Cocktail (Thermo Fisher Cat. No. 78428) and Protease Inhibitor Cocktail (Promega Cat. No. G6521). The following antibodies were used: rabbit anti-pFoxo1 1:1000 (Cat No. 9461, Cell Signaling Technology), rabbit anti-dFoxo1 1:1000 (ab195977, Abcam), rabbit anti-Akt 1:1000 (Cat No. 4691, Cell Signaling Technology), rabbit anti-pAkt 1:1000 (cat No. 4060, Cell Signaling Technology) and mouse anti-actin 1:1000 (DSHB Cat. No. 224236-1). Western blots were developed using Clarify ECL Western Blotting Substrates (BioRad). The bands were detected using an Azure Biosystems c280 digital imager using chemiluminescent detection of HRP. Three independent immunoblots were performed for each experiment.

## Feeding behavior assay
The larval feeding behavior method was adapted from ref. 67. Age-matched third instar larvae (96 hours after egg laying or AEL) were placed in sucrose-agar plates (5% sucrose mixed in 3% agar medium) and allowed to settle for 15 min. The number of mouth-hook contractions per minute were counted. A total number of 30 larvae were scored for each group.

## Generation of germ-free animals
GF animals were generated by following the previously published method[68]. Flies were kept for egg laying in grape juice agar plate for 3-4 hr. Eggs were collected and dechorionated with 2.7% sodium hypochlorite for 2-3 minutes. Dechorionated eggs were washed twice in 70% ethanol and thrice in water, and then transferred to sterile fly food containing tetracycline (50 µg/mL). Flies were reared on sterile food vials for three generations. The absence of bacteria in GF flies was confirmed by 16 S rRNA amplification using the 27 F (5′-AGAGTTT-GATCCTGGCTCAG-3′) and 1492 R (5′-GGTTACCTTGTTACGACTT-3′) primers.

## Growth rate curve
To examine larval growth, we obtained a pool of 20-30 larvae per group and scored their weight at 24, 48, 72, 96, 120, 144, and 168 hours AEL. Average larval weight (mg) was then calculated to generate the growth curve.

## Nile red staining
Midgut and fat body tissues from third-instar larvae were dissected and fixed in 4% paraformaldehyde. Fixation and washing of tissues were followed by the incubation in with 1:2500 dilution of 0.5 mg/mL Nile red (Cat No. 19123, Sigma Aldrich) for half an hour. Upon incubation, tissues were rinsed in water and mounted in 80% glycerol followed by image acquisition. Four animals were used per group. Images were generated as a projection view of 5-10 optical sections. Quantification of Nile red staining in fat body images was done by measuring the red intensity in individual cells from four independent replicates. For comparison between different groups, intensity per cell was divided by the cell area. In midgut images, total red intensity was measured in four independent replicates. For comparison between different groups, intensity was divided by the number of nuclei.

## Triacylglycerol and free fatty acid estimation
Triacylglycerol and free fatty acid levels were estimated from larval midgut and hemolymph. Midgut samples were prepared by homogenizing ~15 midguts in cold 1X PBST (0.05% tween). Hemolymph was collected from 15 larvae and mixed in cold 1X PBST. Triacylglycerol levels were estimated using the manufacturer's protocol (Infinity™ Triglycerides Liquid stable reagent, Thermo scientific #TR22421). Free fatty acid levels were estimated using the manufacturer's protocol (Sigma Aldrich # MAK044). Triacylglycerol and free fatty acid levels were normalized with the protein content determined by Bradford assay.

## Dextran feeding assay and gut barrier defect quantification

Third instar larvae were fed on semi-solid drops of 500 kDa FITC-labelled dextran (Sigma-Aldrich, St. Louis, MO, USA), diluted to 1 mg/mL in sucrose-agar medium (5% sucrose mixed in 3% agar medium) on a petri dish. Larvae were allowed to feed for 15 minutes. Larvae were washed with cold 1X PBS to remove any excess FITC-dextran on the surface. Midguts were dissected out and fixed in 4% paraformaldehyde followed by imaging under a fluorescent microscope. For gut barrier defect quantification, larval guts that failed to retain FITC-signal (green) in their lumen were scored. Pharmacological induction of gut barrier defects was achieved by rearing third instar larvae on food containing 100 μM polyoxin D (Sigma Aldrich # 529313) for 48 hr. Gut barrier defects were examined and quantified using the dextran feeding assay described above.

## Larval midgut sectioning

Intact midguts from age-matched larvae at 96 hours AEL were dissected and fixed in 4% paraformaldehyde, 2% glutaraldehyde, and 0.1 M cacodylic acid. Post-fixation in 2% osmium tetraoxide (OsO4) was followed by dehydration in graded series of alcohol and propylene oxide. The midgut tissues were then embedded and blocked in Embed-812 resin. The resin blocks were sectioned using an ultramicrotome. Thin sections (5 μm) of the anterior midgut regions were obtained and stained with toluidine blue. Images were acquired using light microscopy and presented as greyscale.

## Statistics and reproducibility

Two-tailed unpaired Student's t-test, one-way ANOVA and two-way ANOVA with Šídák's multiple comparisons test, were used for statistical analyses. Significance was ascribed as $P < 0.05$. $P$ values are indicated in all figures. $P$ values falling between 0.99 and 0.0001 are mentioned as exact numbers, while those more than 0.99 or less than 0.0001 are indicated as >0.99 or <0.0001, respectively. Statistical tests and parameters including the sample sizes and the number of independent biological replicates are mentioned in Figure Legends. All experiments, including staining and imaging, were performed at least three times with comparable results. All statistical analyses were performed using GraphPad Prism 9.

## Reporting summary

Further information on research design is available in the Nature Portfolio Reporting Summary linked to this article.

## Data availability

Data are available within the article and supplementary information. Source data are provided as Source Data file with this paper. RNA-seq data generated in this study are available in the National Center for Biotechnology Information Gene Expression Omnibus database under the accession number GSE206229. Source data are provided with this paper.

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

## Acknowledgements
We acknowledge support from the NIH (R35GM130317 to H.J.N.), the Mizutani Foundation for Glycoscience (grant #220061 to H.J.N.), the Grace Science Foundation (research grants to H.J.N. and L.M.S.), AIRC (Investigator grant 20661 to T.V), MUR (PRIN investigator grant 2020CLZ5XW to T.V.) and Telethon Italia (Investigator grant GMR22T1078 to T.V.). We thank the Bloomington *Drosophila* Stock Center (NIH P40OD018537) and the Developmental Studies Hybridoma Bank for reagents; the Microscopy Core of the Baylor College of Medicine (BCM) Intellectual and Developmental Disabilities Research Center (IDDRC) (supported by P50 HD103555 from the Eunice Kennedy Shriver National Institute of Child Health and Human Development [NICHD/NIH]), the BCM Integrated Microscopy Core (supported by NCI-CA125123, NIDDK-56338-13/ 15, and CPRIT-RP150578 and the John S. Dunn Gulf Coast Consortium for Chemical Genomics), Debra Townley for assistance with sectioning midguts for light microscopy, and Professor Pierre Leopold for generously providing *UAS-dilp6* strain and

*Drosophila* anti-Foxo antibody. We also thank members of the Jafar-Nejad and Vaccari labs for discussions.

## Author contributions

A.P., A.G., S.Y.H, T.V., and H.J.N. designed and conceived the project and interpreted. the data. W.F.M., B.S., and L.M.S. performed RNA-seq and transcriptomic analysis. A.P., A.G., S.Y.H., and G.C. performed the experiments. A.P., A.G., T.V., and H.J.N. wrote the manuscript. All authors read, edited, and approved the manuscript.

## Competing interests

The authors declare no competing interests.
