## [Peer Review File · Nature Communications]

Gut barrier defects, intestinal immune hyperactivation and enhanced lipid catabolism drive lethality in NGLY1-deficient *Drosophila*Editorial Note: Parts of this Peer Review File have been redacted as indicated to maintain the confidentiality of unpublished data

REVIEWER COMMENTS

Reviewer #1 (Remarks to the Author):

The manuscript from Pandey et al. aims to describe a role for intestinal PngI in the gut barrier integrity. The authors report that loss PngI leads to increased intestinal immune response, activation of JNK and FOXO pathways thus creating a kind of starvation like state that is supported by the decrease of InR pathway activity and increased lipid catabolism.

This is an original and very interesting study describing for the first time the potential role of PngI in the midgut of *Drosophila* larva. There is only one paper from 2022 (Na, et al. Cytosolic O-GlcNAcylation and PNG1 maintain *Drosophila* gut homeostasis by regulating proliferation and apoptosis. *PLoS Genet.* 18(3): e1010128) that studied the effect of PngI in fly adult midgut, especially on Intestinal Stem Cells.

The authors really try to show a full story linking gut barrier defects and fly lethality due to the absence of PngI. PngI is an enzyme that deglycosylates the denatured form of N-linked glycoproteins in the cytoplasm and assists their proteasome-mediated degradation.

Main comments

- It is not clear in what type of intestinal cells PngI is expressed/required (for preserving PM integrity in PR vs the loss of PngI that leads to increased Immune response in EC vs fat body cells) and how its loss leads to activation of signaling pathways in other cells such as EC.

- Following this first comment, in order to clearly define the tissue that is providing the source of PngI responsible for the phenotypes, it will be very interesting to generate a tissue specific null mutant for PngI.

- Another issue related to the previous point is the fact that, except if I missed something, there is not a single panel showing a complete rescue of the PngI^{-/-} phenotype even the overexpression of PngI (Fig2d, Fig3b-c, Fig4f). For example, according to the model of the authors provided in Fig 7, an overexpression of PngI should increase pAKT. Thus, that does not really strengthen the conclusions of the authors.

For the Fig4, it will be very powerful to show that the overexpression of PngI in the midgut rescues the defect of PM organization.

Could the authors comment the fact that when PngI is overexpressed with Path-Gal4 there is only around 20% of animals with gut barrier defects (Fig4f) while 90% of them are dying (Fig4f).

In Fig 7, it would be very elegant and conclusive to show images of the gut barrier defects rescued by the feeding with HFD.

- The authors described a starvation-like state for the PngI^{-/-} larvae. Since the authors show perturbations of Insulin pathway and differences in body weights (Fig3), we can wonder what is the impact of this null mutation of PngI on the larval growth period. It would be very interesting if the authors could add measurements concerning the growth rate of the animals. That will also improve the clarity the results since the authors do not mention at what stage or what day after egg laying they dissect animals. Do they always use age-matched or stage-matched larvae ?

- The previous comment also highlights that some experimental informations are missing such as the number of replicates, if the graph are representative of one experiment or not, the stage of larvae... It is also not clear what the authors use as "the expected ratio". This is related to the number of pupae obtained from a cross of 5 males and 5 females laying eggs for 3 days. How do the authors normalize their results since they do not count the number of eggs deposited ? Could the authors explain this with more details in Mat and Met ?

For example, in Fig 3F this is not clear whether the larvae are on the same developmental stage. It seems that the anterior spiracles are not at the same developmental stage for PngI^{-/-}, foxo^{+/+}.

In Fig 4f, the path-Gal4 is not reported in the Mat and Met and not clearly defined as well in the text.

In Bloomington, path-Gal4 is reported to drive expression in "GAL4 expressed in embryonic proventriculus, larval brain, eye disc, gut, fat body, adult ovarian squamous and post. terminal follicle cells, male access. glands, seminal vesicle, ejaculatory duct, cyst cells and spermatocytes." Could the authors provide experiments or references to illustrate the expression profile of this driver ? Indeed,

the authors wrote "We found that the penetrance of gut barrier phenotype in Pngl mutant larvae was reduced from 55% to ~20% upon PR-specific overexpression of Pngl (Fig. 4f)" and they conclude that "Pngl is required in PR cells to ensure the integrity of the PM and gut barrier and to promote the survival of Drosophila larvae."

The authors do not well described how they obtain the images from Fig4e. Could they add additional experimental details in the Mat and Met, for example how the samples were prepared, from what part of the midgut, at what larval stage ?

- Could the authors provide some informations concerning the main bacterial species present in their conventionally raised flies ?

Line 483 "infectious insults are not the primary mediators of hyperactive immune response". In this study, the authors work with conventionally raised flies against GF, meaning commensal bacteria and not pathogens. Thus, talking about infectious versus non-infectious responses is an overinterpretation in this case.

Minor comments :

Naming the Germ-free animals as GF, the authors will win space and clarity.

In the text and in the figures, the authors named the yw flies as wild type flies, these are not wild type it is better to rename them control flies.

Line 363 : the authors cannot use the term "gut dysbiosis" here because they use their conventionally raised flies and they did not show that the composition of microbiota is disturbed between yw or PngI-/+ and the mutant PngI-/-.

- Is the phenotype of PM is the same all along the midgut of PngI-/- animals ? Is there a reason to look specially at the proventriculus ?

In the Fig 4h and 4i, the authors show the staining in the "top" proventricular region while in the rest of the study they show midgut images (more often from the anterior part ?). Also, to strengthen the observation of differences in terms of puncta number, the authors should quantify their number, size and localization along the midgut.

By comparing Fig1C and 5D, Edin is not induced anymore, could the authors comment that result ?

In Fig6, by feeding animals with polyoxin D, the authors compare the percentage of gut barrier defects but does feeding with PolyD phenocopies PngI-/- disorganization of PM ? Could the authors provide an image ?

In the Fig7a, could the authors be more precise and whether the images were taken as stacks or snaps ? and how do they quantify the Nile Red staining ? What part of the midgut is shown ?

Reviewer #2 (Remarks to the Author):

I am reviewing Pandey et al. titled "Gut barrier defects, increased intestinal innate immune response, and enhanced lipid catabolism drive lethality in N-glycanase 1 deficient Drosophila". Overall, this is a since study with very thorough experimentation. Conclusions are well supported by results.

1) Is the Pngl-/- the Pnglex14 genotype? If so, please indicate this at the beginning of the results sections. There are a number of published Pngl null alleles.

2) Please add an explanation to the text as to why you are analyzing only L3 larvae

3) Line 107 – In the methods, you say you use an FDR correction for the RNAseq analyses but throughout the paper you say “p value”. And FDR correct P value is normally referred to as a “q value”. Please change this throughout to reflect the actual analyses that you performed.

4) Please include all the RNAseq data in the supplement. Either as individual tabs in a single excel or separate excel files:

a. Full RNAseq data for each of the 3 comparisons (counts, fold change, nominal P, corrected q values, etc). Full data without just the cutoffs is important for the readers to evaluate the data, without having to reanalyze the raw data.

b. A list that reflects each venn diagram in Figure 1A. Lists that include genes unique to each comparison, overlapping 2 comparisons, overlapping all 3 comparisons.

5) It's a little concerning that there are so many unique genes to each comparison. Are they different genetic backgrounds? If they are different genetic backgrounds, which is the best control? Is it simply because of the p value cutoff (some are just miss the cutoff)? There needs to be a little discussion about this. A priori, I wouldn't have thought there would be so many differences between the control groups.

6) Related to point 5, please include a PCA plot. This will help the reader understand how different these samples really are.

7) Why did you use such a conservative FDR of 0.01? Because you got many more genes than expected? Related to point 5 – this conservative FDR make it even more surprising that you got so many genes.

8) Are the guts from males or females or mixed? If mixed, could this be a source of variation in the analyses? Please include this in the text

9) How many replicates per genotype did you sequence? Please add this to the text

10) To avoid confusion, please refer to NFE2L1 and NFE2L2 as NFE2L1/NRF1 and NFE2L2/NRF2

11) The discussion would be improved by placing your results in the context of other studies. For example, there are number of modifier studies in mice and Drosophila, RNAseq data from Drosophila, mice and cells, etc. Are your observations regarding immunity genes, etc novel? Is there evidence from modifiers or other expression studies that suggested this was occurring? Please also comment on whether the immune activation might be occurring in other tissues.

12) Figure 1B should also include fold enrichment of each category. Does this include the full analyses? Please indicate this in the legend and text

13) Please include a supplemental file of the GO analyses that includes categories, fold enrichment, pvalues and importantly, which genes are in each category.

14) Figure 2B, please place the Y axis labels on the Y axis, rather than as a title. This is confusing. A number of figures are like this throughout the paper, please change them all for clarity and consistency

15) Figure 6f and g– it's nearly impossible to see the pJNK signal and IHC image doesn't seem to reflect the quantification.

Reviewer #3 (Remarks to the Author):

In this manuscript, Pandey et al. examine gut barrier dysfunction upon the loss of N-glycanase-1 (Pngl) in the *Drosophila* larval midgut. The authors have elegantly shown that loss of Pngl causes gut barrier defects, leading to reduced insulin signaling and activation of stress-induced JNK signaling. These changes result in the overactivation of Foxo, which leads to hyperactivation of intestinal immune response and increased lipid catabolism, eventually leading to lethality. Additionally, the authors found that growing Pngl mutants under germ-free conditions did not rescue the associated lethality. Instead, rearing the Pngl mutants on an isocaloric high-fat diet reversed the development delay, Foxo overactivation, and animal survival.

This is an exciting and clearly written manuscript. The results are well organized, and the data support the interpretation/conclusions drawn. The data presented in this manuscript have a broad implication and will interest researchers in the ERAD and innate barriers field.

Major:

1. In Fig.4d, using lectin staining (HPA), the authors show that PM is highly disorganized and collapsed on itself, which is quite striking. Do the authors see the collapsed PM phenotype along the entire length of the midgut? The collapsed phenotype observed could also be due to the absence of food bolus or loss in that midgut region during immunostaining. The reduced ectoperitrophic space in the mutants is quite interesting in Fig.4e. Do the authors notice any difference in the thickness of the PM? It would be interesting to assess the PM abnormality at the Ultrastructural level, and such observations will strengthen and complement the PM permeability phenotype in the mutants. In fig4.h-i, the authors found WGA+ and ConA+ intracellular puncta in the PR cells; without co-staining with ER markers, it is premature to conclude that trafficking/secretion of N-glycoproteins is affected. Do the authors find any evidence of ER stress in the PR cells?

Minor:

1. In the introduction few lines about the ERAD pathway would be quite helpful

2. Line 128: The addition of a genomic copy of Pngl fully rescued the immune gene expression. The transcript levels of AMPs and PGRPs in the Pngl[-/-], Pngl Dp/+ background have dropped significantly compared to Pngl[-/-]. Perhaps the authors can rephrase "fully rescued" as it can often imply transcript levels comparable to the control sample

3. Line: 265: Moreover, we observed a significant decrease in the expression of InR in the midgut of Pngl mutants (Fig. 3l). This data can be presented earlier at line:245 after Fig.3i to show a strong case/need to overexpress insulin receptor.

4. Error bars are missing from the control sample in Fig.1c, Supp.Fig.1, Sup.Fig.2a, Fig.2a and 2e, Fig.3e, 3i and 3n, Fig.5d, Fig.6c and 6i and Supp.Fig.4c

5. In the transcriptomic analysis, there are several putative components of the PM (Muc26B, Muc96D..) that are downregulated, and it is worth discussing their role in the PM function.

6. Line:308, the authors observed that the midgut displayed a dense lumen content (potentially due to the gut clearance defect previously reported), Its worth mentioning in the discussion about the involvement of enteric neurons in regulating gut peristalsis and nutrient sensing and regulatory peptide secretion by the enteroendocrine cells of the midgut under starvation conditions.

Kenmoku H., Ishikawa H., Ote M., Kuraishi T., Kurata S. A subset of neurons controls the permeability of the peritrophic matrix and midgut structure in *Drosophila* adults. *J. Exp. Biol.* 2016;219:2331–2339. doi: 10.1242/jeb.122960.

Siviter R.J., Coast G.M., Winther A.M., Nachman R.J., Taylor C.A., Shirras A.D., Coates D., Isaac R.E., Nassel D.R. Expression and functional characterization of a *Drosophila* neuropeptide precursor with homology to mammalian preprotachykinin A. *J. Biol. Chem.* 2000;275:23273–23280. doi: 10.1074/jbc.M002875200

Control of lipid metabolism by tachykinin in *Drosophila*. *Cell Rep.* 2014;9:40–47. doi: 10.1016/j.celrep.2014.08.060.

Reviewer #1 (Remarks to the Author):

The manuscript from Pandey et al. aims to describe a role for intestinal Pngl in the gut barrier integrity. The authors report that loss Pngl leads to increased intestinal immune response, activation of JNK and FOXO pathways thus creating a kind of starvation like state that is supported by the decrease of InR pathway activity and increased lipid catabolism.

This is an original and very interesting study describing for the first time the potential role of Pngl in the midgut of Drosophila larva. There is only one paper from 2022 (Na, et al. Cytosolic O-GlcNAcylation and PNG1 maintain Drosophila gut homeostasis by regulating proliferation and apoptosis. PLoS Genet. 18(3): e1010128) that studied the effect of Pngl in fly adult midgut, especially on Intestinal Stem Cells. The authors really try to show a full story linking gut barrier defects and fly lethality due to the absence of Pngl. Pngl is an enzyme that deglycosylates the denatured form of N-linked glycoproteins in the cytoplasm and assists their proteasome-mediated degradation.

We thank the reviewer for the positive evaluation of the manuscript.

Main comments

- It is not clear in what type of intestinal cells Pngl is expressed/required (for preserving PM integrity in PR vs the loss of Pngl that leads to increased Immune response in EC vs fat body cells) and how its loss leads to activation of signaling pathways in other cells such as EC.

- Following this first comment, in order to clearly define the tissue that is providing the source of Pngl responsible for the phenotypes, it will be very interesting to generate a tissue specific null mutant for Pngl.

The data presented in the first submission of the manuscript suggest that loss of *Pngl* in several cell types including PR cells, enterocytes, and fat body contributes to increased immune gene expression in the midgut and lethality. We showed that loss of *Pngl* in PR cells (present in proventriculus) is associated with PM defects. However, loss of *Pngl* in PR cells, enterocytes and fat body all contribute to immune gene induction in the midgut and lethality. Having said that, the reviewer's comment reminded us that we have not examined the other two cell types in the larval midgut epithelium: enteroendocrine cells, and adult midgut precursor cells.

To answer the reviewer's query about the intestinal cell type, we considered either performing clonal analysis or RNAi-based knockdown of *Pngl*. *Pngl* is located on chromosome 2R at 42A7-42A8. Since the gene is between the centromere and the available FRT insertions on 2R (42B and 42D), we were not able to perform classical clonal analysis on *Pngl*. Therefore, we used a previously published *UAS-Pngl-RNAi* line (Galeone et al., 2017) for cell type-specific knockdown of *Pngl*. Our data (shown in revised Fig. 2c and 2d) indicate that among these three cell types (enterocytes, enteroendocrine cells, and adult midgut precursor cells) only enterocyte-specific *Pngl* knockdown results in significant lethality and immune gene induction. Moreover, *Pngl* knockdown in none of these cell types led to gut barrier defect (data shown below for the reviewer; Rebuttal Fig 1), further supporting the conclusion that the gut barrier defect in *Pngl* mutant larvae are caused by loss of *Pngl* in the PR cells.

Rebuttal Fig. 1. Graph showing quantification of gut barrier defects in the indicated genotypes. Note that only *Path-GAL4* driven *Pngl* knockdown shows gut barrier defects.

- Another issue related to the previous point is the fact that, except if I missed something, there is not a single panel showing a complete rescue of the *Pngl*^{-/-} phenotype even the overexpression of *Pngl* (Fig2d, Fig3b-c, Fig4f). For example, according to the model of the authors provided in Fig 7, an overexpression of *Pngl* should increase pAKT. Thus, that does not really strengthen the conclusions of the authors.

We have previously shown that the lethality of *Pngl* homozygous mutants can be fully rescued by one copy of a *Pngl* genomic duplication (Han et al., 2020) and by ubiquitous overexpression of the human NGLY1 (Galeone et al., 2017). Overexpression of NGLY1 with mesodermal drivers rescued the lethality of *Pngl* mutants up to 80%, indicating that *Pngl* plays important roles in other cell types as well. In the current manuscript, we provide evidence that in addition to its important role in visceral mesoderm, *Pngl* is also required in at least three other cell types to promote survival in *Drosophila* larvae: enterocytes, fat body, and the PR cells. Together, our previous work and current manuscript indicate that no single cell type can explain the whole lethality and phenotypes of *Pngl* mutants. Therefore, overexpression of *Pngl* in one cell type does not rescue them all. In the revised manuscript (including the abstract), we have tried to better describe the conclusion that *Pngl* is required in multiple cell types to promote larval development and survival.

As per the reviewer's comment, in the revised manuscript, we have also included new rescue data by a *Pngl* genomic duplication for Foxo overactivation (revised Fig 3a, b), decreased Akt phosphorylation (revised Fig. 3c, f), and intracellular puncta in PR cells (revised Fig 4h). In addition, we now show that enterocyte-specific overexpression of *Pngl* rescues pAkt levels in *Pngl*^{-/-} midguts (revised Fig. 3f). Together with our previous work on the roles of *Pngl* in visceral mesoderm, these data indicate that overexpression *Pngl* in each cell type almost fully rescues the phenotypes related to that cell type, but the animal lethality is caused by combined effects of loss of *Pngl* in various cell types. We have also revised the model figure (Fig. 7g in revised manuscript) to indicate the different cell types involved in the loss of *Pngl* phenotypes included in this study.

For the Fig4, it will be very powerful to show that the overexpression of *Pngl* in the midgut rescues the defect of PM organization.

In *Drosophila* larvae, PM is only secreted from PR cells present in the proventriculus region and then moves continuously towards the midgut region. Our data suggest that PM defects are due to loss of *Pngl* in PR cells. In the first submission, we reported that overexpression of WT *Pngl* in PR cells significantly improves the gut barrier defect in *Pngl* mutant larvae. In the revised version, we have repeated the experiment with a catalytically-inactive version of *Pngl* (*Pngl*^{C303A}) and did not see any rescue of gut barrier defects in *Pngl* mutants (revised Fig. 4f). As per the reviewer's suggestion, we also examined if *Pngl* overexpression in enterocytes can rescue the gut barrier defect but did not see any rescue (data shown below for the reviewer; Rebuttal Fig 2). Please also see our response to another comment by the reviewer related to Path-GAL4 expression pattern.

Rebuttal Fig. 2. Graph showing the quantification of gut barrier defects upon enterocyte specific overexpression of *Pngl* in *Pngl* mutants.

Could the authors comment the fact that when *Pngl* is overexpressed with Path-Gal4 there is only around 20% of animals with gut barrier defects (Fig4f) while 90% of them are dying (Fig4f).

Our data suggest that the lethality in *Pngl* mutants is a combined outcome of loss of *Pngl* in different cell types. As described in response to a previous question (second comment) by the reviewer, our previous and current work indicate that *Pngl* functions in multiple cell types to promote full survival of *Drosophila* larvae. Loss of *Pngl* in each of these cell types is only partially lethal. We previously established the role of this enzyme in visceral mesoderm. In the current study, we showed that *Pngl* is also required in enterocytes, fat body, and PM secretory cells (PR cells). Loss of *Pngl* in these cell types converges onto Foxo overactivation, subsequent hyperactive immune gene expression, and enhanced lipid catabolism leading to lethality. Therefore, although PR cell-specific *Pngl* overexpression significantly rescues the gut barrier phenotype it only partially rescues the lethality, as it cannot improve the detrimental effects of loss of *Pngl* in other cell types (visceral mesoderm, enterocytes, and the fat body).

In Fig 7, it would be very elegant and conclusive to show images of the gut barrier defects rescued by the feeding with HFD.

To address this point, we examined the gut barrier dysfunction using a FITC-dextran feeding assay in animals raised on standard versus HFD. We scored ~45 *Pngl* mutant larvae (3 cohorts of 15 animals each) fed on a standard diet and ~60 *Pngl* mutant larvae (3 cohorts of 20 animals each) fed on HFD. We did not see any rescue of barrier defects in *Pngl* mutants fed on HFD (data shown below; Rebuttal Fig 3), indicating that HFD does not correct the gut barrier defect. Our data suggest that the gut barrier defect is caused by *Pngl* loss in PR cells (potentially a secretory defect in *Pngl* mutant PR cells). Moreover, HFD is likely bypassing the gut barrier defects by providing sufficient energy to *Pngl* mutants, as it rescues

Foxo activation, immune gene expression, and lethality in these animals. Therefore, the observation that HFD does not rescue the gut barrier defect matches our other data and model.

Rebuttal Fig. 3. Graph showing quantification of gut barrier defects in standard- and high-fat diet-fed *Pngl* mutants.

- The authors described a starvation-like state for the *Pngl*^{-/-} larvae. Since the authors show perturbations of Insulin pathway and differences in body weights (Fig3), we can wonder what is the impact of this null mutation of *Pngl* on the larval growth period. It would be very interesting if the authors could add measurements concerning the growth rate of the animals. That will also improve the clarity the results since the authors do not mention at what stage or what day after egg laying they dissect animals. Do they always use age-matched or stage-matched larvae ?

Since *Pngl* mutants show developmental delay, we used age-matched animals for the study. We have incorporated the details in the revised manuscript. Moreover, as suggested by the reviewer, we have compared the growth rate of mutant and control animals and have provided the data in the revised Fig. 3g.

- The previous comment also highlights that some experimental informations are missing such as the number of replicates, if the graph are representative of one experiment or not, the stage of larvae... It is also not clear what the authors use as “the expected ratio”. This is related to the number of pupae obtained from a cross of 5 males and 5 females laying eggs for 3 days. How do the authors normalize their results since they do not count the number of eggs deposited ? Could the authors explain this with more details in Mat and Met ?

We have now added the details in “Materials and Methods” section in the revised manuscript. Also, prompted by this comment and a comment by Reviewer 3, we have reanalyzed many of our quantifications and provided additional information about the data in most of our graphs.

For example, in Fig 3F this is not clear whether the larvae are on the same developmental stage. It seems that the anterior spiracles are not at the same developmental stage for *Pngl*^{-/-}, *foxo*^{+/+}. We thank the reviewer for raising this point. We have used age-matched animals ~96hr AEL. We have added better images with better resolution. We have added the age details in the corresponding figure legend as well.

In Fig 4f, the path-Gal4 is not reported in the Mat and Met and not clearly defined as well in the text. In Bloomington, path-Gal4 is reported to drive expression in “GAL4 expressed in embryonic proventriculus, larval brain, eye disc, gut, fat body, adult ovarian squamous and post. terminal follicle

cells, male access. glands, seminal vesicle, ejaculatory duct, cyst cells and spermatocytes.” Could the authors provide experiments or references to illustrate the expression profile of this driver? Indeed, the authors wrote “We found that the penetrance of gut barrier phenotype in *Pngl* mutant larvae was reduced from 55% to ~20% upon PR-specific overexpression of *Pngl* (Fig. 4f)” and they conclude that “*Pngl* is required in PR cells to ensure the “integrity of the PM and gut barrier and to promote the survival of *Drosophila* larvae.”

We thank the reviewer for the comment. We examined the expression pattern of *Path-GAL4* in larvae using *UAS-mCherry^{nls}* (*Path-GAL4>UAS-mCherry^{nls}*) and found that as the reviewer mentioned, in addition to the proventriculus, *Path-GAL4* also drives expression in the larval brain, salivary gland, and fat body (shown in revised supplementary Fig. 3a). In the first submission, we showed that overexpression of *Pngl* using *Path-GAL4* significantly reduced the gut barrier defects in *Pngl* mutant larvae (original Fig. 4f). To ask whether overexpression of *Pngl* driven in other tissues by *Path-GAL4* (larval brain, salivary gland, and fat body) had any role in the observed rescue, we overexpressed *Pngl* using GAL4 drivers specific to these tissues and examined the gut barrier in *Pngl* mutants. Overexpression of *Pngl* in none of these tissues rescued the gut barrier defects in *Pngl* mutants (shown in revised supplementary Fig. 3b). Moreover, knockdown of *Pngl* in these tissues (fat body, brain, salivary gland) did not show a gut barrier phenotype (shown in revised supplementary Fig. 3c). Therefore, we conclude that *Pngl* is required in PR cells to ensure the integrity of the PM and gut barrier. These data (shown in revised Supplementary Fig.3) combined with the new data in the revised Fig. 4f and Fig. 4g demonstrate that the enzymatic activity of *Pngl* is specifically required in PR cells to promote gut barrier integrity and survival of larvae.

The authors do not well described how they obtain the images from Fig4e. Could they add additional experimental details in the Mat and Met, for example how the samples were prepared, from what part of the midgut, at what larval stage ?

In the revised manuscript, we have added the information to the “Materials and Methods” section. Midguts from the larvae (96hr AEL) were dissected and fixed in paraformaldehyde, glutaraldehyde, and cacodylic acid. Following post-fixation in osmium tetroxide, tissues were dehydrated through a graded series of alcohol and propylene oxide, and then embedded in epoxy resin. Resin blocks were then sectioned with an ultramicrotome, stained with toluidine blue and presented as grayscale images.

- Could the authors provide some informations concerning the main bacterial species present in their conventionally raised flies ?

[redacted]

Line 483 “infectious insults are not the primary mediators of hyperactive immune response”. In this study, the authors work with conventionally raised flies against GF, meaning commensal bacteria and not pathogens. Thus, talking about infectious versus non-infectious responses is an overinterpretation in this case.

We agree with the reviewer. Since we have only removed the bacteria (by making germ-free animals), using “infection/non-infectious” can be considered over-interpretation. To stay close to data, we have replaced the word “infectious” with more neutral alternatives like “gut bacteria versus non-bacterial insults” throughout the manuscript.

Minor comments :

Naming the Germ-free animals as GF, the authors will win space and clarity.

Done.

In the text and in the figures, the authors named the yw flies as wild type flies, these are not wild type it is better to rename them control flies.

That’s correct. Done.

Line 363 : the authors cannot use the term “gut dysbiosis” here because they use their conventionally raised flies and they did not show that the composition of microbiota is disturbed between yw or *Pngl*^{+/+} and the mutant *Pngl*^{-/-}.

We agree. We have replaced “gut dysbiosis” with “the gut microbiota”.

- Is the phenotype of PM is the same all along the midgut of *Pngl*^{-/-} animals ? Is there a reason to look specially at the proventriculus ?

The PM phenotypes observed in *Pngl*-mutant larvae include PM disorganization, collapsing, and close proximity of PM with gut epithelium. We observed that the PM collapsing is mostly in the anterior midgut, while the posterior midgut exhibits loss of peritrophic space (leading to the close proximity of PM and gut epithelium). In *Drosophila* larvae, PM is exclusively secreted from the PR cells which are present in the proventriculus region. Therefore, we have looked at the proventriculus region to examine any possible defect with PR cells.

In the Fig 4h and 4i, the authors show the staining in the “top” proventricular region while in the rest of the study they show midgut images (more often from the anterior part ?). Also, to strengthen the observation of differences in terms of puncta number, the authors should quantify their number, size and localization along the midgut.

In *Drosophila* larvae, PM is only secreted from the PR cells. Therefore, to examine if PM defects in *Pngl* mutants are associated with any defects in PM secretory cells (i.e., PR cells), we performed lectin staining in the proventriculus region. To address the reviewer’s comments, we added a third genotype to our analysis (*Pngl*^{-/-} rescued by a *Pngl* genomic duplication) and quantified the number and size of the puncta. In the revised manuscript, we have added the rescue images and the puncta number and size quantification (revised Fig. 4h-j).

By comparing Fig1C and 5D, *Edin* is not induced anymore, could the authors comment that result ?

In original Fig. 1c (Fig. 1d in the revised manuscript) animals are conventionally reared, while in Fig. 5d, germ-free animals were used. Our data suggest that induction of *edin* is in response to bacteria in *Pngl* mutant larvae and therefore it is not increased in the midgut of germ-free animals. This is in line with a previous report that expression of *edin* (elevated during infection) depends on the Imd pathway transcription factor Relish (Vanha-aho, Plos One, 2012).

In Fig6, by feeding animals with polyoxin D, the authors compare the percentage of gut barrier defects but does feeding with PolyD phenocopies *Pngl*^{-/-} disorganization of PM ? Could the authors provide an image ?

Our data indicate that impairment of the gut barrier function, potentially resulting from PM disorganization, contributes to immune gene activation and lethality in *Pngl* mutants. Poly D is a known inhibitor of chitin synthase and thereby impairs PM formation in insects. We had shown a quantification of gut barrier defects caused by Poly D in the original manuscript and have now provided images of Poly D-fed larvae raised on 500 kDa FITC-dextran in the revised Fig. 6a. PolyD-fed animals also showed immune gene induction and mild lethality phenotype. Accordingly, although the mechanism of PM abnormality might not be the same in Poly D-fed versus *Pngl*-mutant larvae, both of these conditions result in gut barrier dysfunction. Therefore, we utilized Poly D feeding as a pharmacological approach to recapitulate the gut barrier defect observed in *Pngl* mutants.

In the Fig7a, could the authors be more precise and whether the images were taken as stacks or snaps ? and how do they quantify the Nile Red staining ? What part of the midgut is shown ?

In the revised version, we have added detailed information in the “Materials and Methods” section to address this point. Both fat body and midgut images are projections of 5-10 optical sections covering the Nile red-positive parts of the tissues. For the fat body, quantification is based on manual measurement of color intensity (using image J) in individual cells normalized to the area. For the midguts, total color intensity was normalized to the number of nuclei in each image. For each group, 4-5 replicates were used. Nile Red staining was performed on the anterior midgut for each group.

Reviewer #2 (Remarks to the Author):

I am reviewing Pandey et al. titled “Gut barrier defects, increased intestinal innate immune response, and enhanced lipid catabolism drive lethality in N-glycanase 1 deficient *Drosophila*”. Overall, this is a since study with very thorough experimentation. Conclusions are well supported by results.

We thank the reviewer for the positive evaluation of our manuscript.

1) Is the *Pngl*^{-/-} the *Pnglex14* genotype? If so, please indicate this at the beginning of the results sections. There are a number of published *Pngl* null alleles.

Thanks for raising this point. Yes, *Pngl*^{-/-} and *Pngl*^{ex14/ex14} are the same genotypes. We have added this in the beginning of the Results section in the revised manuscript.

2) Please add an explanation to the text as to why you are analyzing only L3 larvae

Pngl mutants display developmental delay, with the majority of the mutant animals not reaching the pupal stage (Galeone et al., 2017; eLife). Therefore, consistent with the previously published work from our lab using *Pngl* mutants (Galeone et al., eLife, 2017; Galeone et al., eLife, 2020; Han et al, PLoS Genetics, 2020), we used third instar larvae in this study. We have included this explanation in the revised manuscript.

3) Line 107 – In the methods, you say you use an FDR correction for the RNAseq analyses but throughout the paper you say “p value”. And FDR correct P value is normally referred to as a “q value”. Please change this throughout to reflect the actual analyses that you performed.

We thank the reviewer for bringing this issue to our attention. As described in the revised manuscript, the p-values were corrected for multiple testing using the Benjamini-Hochberg approach. Although this method is related to the q-value produced through the Storey-Tibshirani method, it is not exactly the same. Therefore, we have changed P-value to the term FDR throughout the revised manuscript to avoid any reader confusion.

4) Please include all the RNAseq data in the supplement. Either as individual tabs in a single excel or separate excel files:

a. Full RNAseq data for each of the 3 comparisons (counts, fold change, nominal P, corrected q values, etc). Full data without just the cutoffs is important for the readers to evaluate the data, without having to reanalyze the raw data.

These data are included in the new Table S1 in the revised manuscript.

b. A list that reflects each venn diagram in Figure 1A. Lists that include genes unique to each comparison, overlapping 2 comparisons, overlapping all 3 comparisons.

To address this comment, we recreated the Shiny app that was originally used to generate the differentially expressed gene list in the manuscript and have provided the web address for this Shiny app in the revised manuscript. The app is hosted by EMBL in a permanent location. Using this app, the readers of our manuscript can change the parameters if they wish and generate various comparison lists at different significance levels and can even select the gene types (protein coding, ncRNA, snRNA, etc.).

5) It’s a little concerning that there are so many unique genes to each comparison. Are they different genetic backgrounds? If they are different genetic backgrounds, which is the best control? Is it simply because of the p value cutoff (some are just miss the cutoff)? There needs to be a little discussion about this. A priori, I wouldn’t have thought there would be so many differences between the control groups.

The genetic background of Pngl[+/-] and Pngl[+/-];PnglDp are closer to the Pngl[-/-] compared to y w. However, Pngl[+/-] and Pngl[+/-]; PnglDp have one functional copy of Pngl, unlike y w which has two. The majority of the studies in the literature compare the mutant strains with one control. To be more stringent, we have used three different controls and focused on the overlap of all three pairwise comparisons of Pngl[-/-] animals. Importantly, so far qRT-PCR experiments have confirmed the genes from the RNAseq that we have tested.

6) Related to point 5, please include a PCA plot. This will help the reader understand how different these samples really are.

We have added a PCA plot as Fig. 1a in the revised manuscript. The PCA plot shows the first two first principal components driving the majority of the variance. The plot was generated using the log-transformed counts-per-million (CPM). All samples show strong distinctive clustering by genotype.

7) Why did you use such a conservative FDR of 0.01? Because you got many more genes than expected? Related to point 5 – this conservative FDR make it even more surprising that you got so

many genes.

We compared FDR 0.05 and FDR 0.01 and found that the major differentially expressed categories were similar for these two FDRs. Moreover, by switching the FDR from 0.01 to 0.05, there were only minor changes in the number of differentially expressed genes. Therefore, in the revised manuscript we have used $FDR \leq 0.05$. The minor changes observed upon switching from 0.01 to 0.05 further indicate that focusing on the overlap of all three pairwise comparisons has made our strategy highly stringent. Importantly, the upregulated gene categories on which we focus in the manuscript (immune genes and lipases) are both confirmed by qRT-PCR.

8) Are the guts from males or females or mixed? If mixed, could this be a source of variation in the analyses? Please include this in the text

Thanks for raising this point. We have used a mixed pool of males and female samples for RNAseq experiments. We have included this information in the revised manuscript. We agree with the reviewer that this issue can be a source of variation in our RNA-seq data. However, the immune genes and lipases were increased in Pngl mutants based on all three pair-wise comparisons.

9) How many replicates per genotype did you sequence? Please add this to the text

We have used two biological replicates for RNAseq analysis and this information has now been added to the revised manuscript (also shown in the PCA plot as Fig 1a in the revised manuscript).

10) To avoid confusion, please refer to NFE2L1 and NFE2L2 as NFE2L1/NRF1 and NFE2L2/NRF2
Done.

11) The discussion would be improved by placing your results in the context of other studies. For example, there are number of modifier studies in mice and Drosophila, RNAseq data from Drosophila, mice and cells, etc. Are your observations regarding immunity genes, etc novel? Is there evidence from modifiers or other expression studies that suggested this was occurring? Please also comment on whether the immune activation might be occurring in other tissues.

An increase in immune genes was observed by others in transcriptome profiling of NGLY1 deficiency patient fibroblasts and in adult flies with RNAi-mediated Pngl knockdown (Rauscher et al, Biochem J, 2022; Owings et al, HMG, 2018). Therefore, although we have not tested this, immune activation is likely to occur in other tissues as well. We have revised the last paragraph of the Discussion to address this point and to cite these two references. Both references were already cited elsewhere in the original manuscript.

12) Figure 1B should also include fold enrichment of each category. Does this include the full analyses? Please indicate this in the legend and text

We have revised this figure by providing fold enrichment and explaining in legend as per the reviewer's suggestion. Fold enrichments are also included in the revised Supplementary Table 3.

13) Please include a supplemental file of the GO analyses that includes categories, fold enrichment, p values and importantly, which genes are in each category.

In the revised manuscript, we have included Supplementary Table 3 with the list of top 15 biological processes identified upon GO analysis of the upregulated and downregulated gene categories, along with gene count, fold enrichment, *p* value, FDR, and the gene list for each category.

14) Figure 2B, please place the Y axis labels on the Y axis, rather than as a title. This is confusing. A number of figures are like this throughout the paper, please change them all for clarity and consistency

Done.

15) Figure 6f and g– it's nearly impossible to see the pJNK signal and IHC image doesn't seem to reflect the quantification.

We have replaced the images with better exposure and have revised the quantification.

Reviewer #3 (Remarks to the Author):

In this manuscript, Pandey et al. examine gut barrier dysfunction upon the loss of N-glycanase-1 (Pngl) in the *Drosophila* larval midgut. The authors have elegantly shown that loss of Pngl causes gut barrier defects, leading to reduced insulin signaling and activation of stress-induced JNK signaling. These changes result in the overactivation of Foxo, which leads to hyperactivation of intestinal immune response and increased lipid catabolism, eventually leading to lethality. Additionally, the authors found that growing Pngl mutants under germ-free conditions did not rescue the associated lethality. Instead, rearing the Pngl mutants on an isocaloric high-fat diet reversed the development delay, Foxo overactivation, and animal survival.

This is an exciting and clearly written manuscript. The results are well organized, and the data support the interpretation/conclusions drawn. The data presented in this manuscript have a broad implication and will interest researchers in the ERAD and innate barriers field.

We thank the reviewer for the positive evaluation of our manuscript.

Major:

1. In Fig.4d, using lectin staining (HPA), the authors show that PM is highly disorganized and collapsed on itself, which is quite striking. Do the authors see the collapsed PM phenotype along the entire length of the midgut? The collapsed phenotype observed could also be due to the absence of food bolus or loss in that midgut region during immunostaining.

PM defects in *Pngl* mutants include disorganization, collapsing, and close proximity with enterocytes (loss of peritrophic space). We observe the collapsing phenotype mainly in the anterior midgut. However, loss of peritrophic space is more frequent towards the posterior end of the midgut. To address the reviewer's concern if the PM collapse is due to the absence of food, we tried lectin staining on FITC-dextran-fed mutant larvae, but unfortunately we lose the FITC-dextran signal during the lectin staining and processing steps. However, we have previously reported that Pngl mutants have a severe gut clearance defect and accumulate food in the midgut, arguing against the absence of food bolus. Moreover, we did not find the PM collapsing phenotype in the control larvae (scored over 30 animals in

different sets of experiments), suggesting that the observed lectin staining pattern is not a staining artefact and that there is PM collapsing in *Pngl* mutants. We note that the gut barrier defect seems to drive some of the phenotypes in *Pngl* mutants, and this phenotype is confirmed by FITC-dextran experiments independently of the PM collapse data.

The reduced ectoperitrophic space in the mutants is quite interesting in Fig.4e. Do the authors notice any difference in the thickness of the PM? It would be interesting to assess the PM abnormality at the Ultrastructural level, and such observations will strengthen and complement the PM permeability phenotype in the mutants.

[redacted]

In fig4.h-i, the authors found WGA+ and ConA+ intracellular puncta in the PR cells; without co-staining with ER markers, it is premature to conclude that trafficking/secretion of N-glycoproteins is affected. Do the authors find any evidence of ER stress in the PR cells?

To address this issue, we performed co-staining for ConA and the ER marker Calnexin 99 (Cnx99A). Unfortunately, as shown in the following figure for the reviewer (Rebuttal Fig. 5a), the data are not conclusive, as we see broad distribution of the ER marker in PR cells and cannot say objectively whether most ConA+ puncta are in the ER or not. New data in revised Fig. 4h and 4i show that addition of *Pngl*

Dp rescues the intracellular puncta in *Pngl* mutants, proving that these puncta appear due to the loss of Pngl. We have also performed additional rescue experiments by overexpressing WT Pngl or a catalytically-inactive Pngl (Pngl^{C303A}) with Path-GAL4 in *Pngl*^{-/-} larvae. As shown in the revised Fig. 4k-m, only WT Pngl was able to rescue the number and size of the puncta in PR cells. We also provide new data (revised Fig. 4f and 4g) showing that Pngl^{C303A} is not able to rescue the gut barrier defect and lethality in *Pngl*^{-/-} larvae. Together, these data suggest an important role for the enzymatic activity of Pngl in these cells. However, given the comment by the reviewer and the inconclusive nature of our ER staining shown in Rebuttal Fig. 5a, we have modified the fourth paragraph of the Discussion to acknowledge that “It remains to be studied whether the gut barrier defects observed in *Pngl*^{-/-} larvae result from a failure in the deglycosylation and secretion of key *N*-glycoprotein components of the PM.”

We also dissected larval proventriculi from a number of *y w* and *Pngl*-mutant larvae and performed western blot to detect the ER stress marker BiP (Grp78). As shown in the following figure (Rebuttal Fig 5b), we did not observe an increase in the level of BiP in mutant proventriculus samples, when compared to control samples. This observation suggests that ER stress is not activated in the PR cells and is consistent with previous reports showing no sign of ER stress upon *Pngl* loss (e.g., Owings et al, HMG, 2018). We note that we have previously reported some degree of BiP induction and ER stress in Ngly1-mutant mouse embryonic fibroblasts (Galeone et al, eLife, 2020).

Rebuttal Fig. 5 a, a'. Confocal images showing co-staining for ConA, the ER marker Cnx99A and DAPI. **b.** Western blotting and quantification for BiP in control and *Pngl*-mutant proventriculus samples. BiP levels were not statistically different in the two genotypes.

Minor:

1. In the introduction few lines about the ERAD pathway would be quite helpful

In the revised manuscript, we have added two sentences about ERAD to the Introduction.

2. Line 128: The addition of a genomic copy of Pngl fully rescued the immune gene expression. The transcript levels of AMPs and PGRPs in the *Pngl*^{-/-}, *Pngl* *Dp*/+ background have dropped significantly compared to *Pngl*^{-/-}. Perhaps the authors can rephrase “fully rescued” as it can often

imply transcript levels comparable to the control sample

In the revised manuscript, we have rephrased the sentence to “Addition of a genomic copy of *Pngl* significantly reduced the immune gene expression”.

3. Line: 265: Moreover, we observed a significant decrease in the expression of InR in the midgut of *Pngl* mutants (Fig. 3I). This data can be presented earlier at line:245 after Fig.3i to show a strong case/need to overexpress insulin receptor.

We have moved up the panel and the text highlighted by the reviewer in the revised manuscript.

4. Error bars are missing from the control sample in Fig.1c, Supp.Fig.1, Sup.Fig.2a, Fig.2a and 2e, Fig.3e, 3i and 3n, Fig.5d, Fig.6c and 6i and Supp.Fig.4c

In response to this comment and one of the issues raised by reviewer 1 and based on Nature Communications guidelines for data presentation, we have reanalyzed many of our quantifications and statistical analyses and have added the error bars to all controls in the revised manuscript.

5. In the transcriptomic analysis, there are several putative components of the PM (Muc26B, Muc96D..) that are downregulated, and it is worth discussing their role in the PM function.

Thank you. We have added this idea in the Discussion section of the revised manuscript.

6. Line:308, the authors observed that the midgut displayed a dense lumen content (potentially due to the gut clearance defect previously reported), Its worth mentioning in the discussion about the involvement of enteric neurons in regulating gut peristalsis and nutrient sensing and regulatory peptide secretion by the enteroendocrine cells of the midgut under starvation conditions.

Kenmoku H., Ishikawa H., Ote M., Kuraishi T., Kurata S. A subset of neurons controls the permeability of the peritrophic matrix and midgut structure in *Drosophila* adults. *J. Exp. Biol.* 2016;219:2331–2339. doi: 10.1242/jeb.122960.

Siviter R.J., Coast G.M., Winther A.M., Nachman R.J., Taylor C.A., Shirras A.D., Coates D., Isaac R.E., Nassel D.R. Expression and functional characterization of a *Drosophila* neuropeptide precursor with homology to mammalian preprotachykinin A. *J. Biol. Chem.* 2000;275:23273–23280. doi: 10.1074/jbc.M002875200

Control of lipid metabolism by tachykinin in *Drosophila*. *Cell Rep.* 2014;9:40–47. doi: 10.1016/j.celrep.2014.08.060.

We thank the reviewer for the suggestion. We have added text to the first paragraph of the Discussion in the revised manuscript to acknowledge that “nutrient deprivation can also suppress lipogenesis in enterocytes and the fat body through enhanced secretion of Tachykinins by enteroendocrine cells⁵⁹. It remains to be examined whether reduced lipogenesis also contributes to low lipid content in *Pngl*^{-/-} larvae.” (Reference 59 is the last reference on the list provided by the reviewer). We have previously shown that the peristalsis defect in *Pngl* mutants can be fully explained by reduced AMPK α signaling in the visceral mesoderm and is not driven by loss of *Pngl* in neurons (Han et al, *PLoS Genetics*, 2020). Therefore, the gut clearance defects in *Pngl* mutants are not likely to be related to enteric neurons.

REVIEWERS' COMMENTS

Reviewer #2 (Remarks to the Author):

Thank you for addressing my concerns. The paper is much clearer now. I just have one remaining request and I insist on this. Per my point in 4B. Thank you for including the shiny app link, but I still insist that you include a supp file that has the lists of genes in the venn diagrams. Many readers just want to check the genes and see if their gene of interest shows up. supp list serves this purpose. not all readers need or have time to reanalyze your data just for that point. Please include that table.

Reviewer #3 (Remarks to the Author):

The authors have diligently and satisfactorily responded to all the comments and concerns raised during the review process and have addressed each point by offering well-reasoned explanations and presenting additional data where necessary. Overall, their thorough and thoughtful responses and revised supporting data have successfully addressed all concerns and are sufficient to support their findings and conclusions.

Point-by-point response

Reviewer #2 (Remarks to the Author):

Thank you for addressing my concerns. The paper is much clearer now. I just have one remaining request and I insist on this. Per my point in 4B. Thank you for including the shiny app link, but I still insist that you include a supp file that has the lists of genes in the venn diagrams. Many readers just want to check the genes and see if their gene of interest shows up. supp list serves this purpose. not all readers need or have time to reanalyze your data just for that point. Please include that table.

The requested table is added as new Table S2.

Reviewer #3 (Remarks to the Author):

The authors have diligently and satisfactorily responded to all the comments and concerns raised during the review process and have addressed each point by offering well-reasoned explanations and presenting additional data where necessary. Overall, their thorough and thoughtful responses and revised supporting data have successfully addressed all concerns and are sufficient to support their findings and conclusions.

Thank you.